

# 1  Retrieval of aerosol composition directly from
# 2  satellite and ground-based measurements

Lei Li[1,2], Oleg Dubovik[2*], Yevgeny Derimian[2*], Gregory L Schuster[3], Tatyana
Lapyonok[2], Pavel Litvinov[4], Fabrice Ducos[2], David Fuertes[4], Cheng Chen[2],
Zhengqiang Li[5], Anton Lopatin[4], Benjamin Torres[2] , Huizheng Che[1]
[1]State Key Laboratory of Severe Weather (LASW) and Institute of Atmospheric
Composition, Chinese Academy of Meteorological Sciences, CMA, Beijing, 100081, China
[2]Univ. Lille, CNRS, UMR 8518 - LOA - Laboratoire d'Optique Atmosphérique, F-59000
Lille, France
[3]NASA Langley Research Center, Hampton, VA, USA
[4]GRASP-SAS, Remote Sensing Developments, Cité Scientifique, Univ. Lille, Villeneuve
d'Ascq, 59655, France
[5]State Environmental Protection Key Laboratory of Satellite Remote Sensing, Institute of
Remote Sensing and Digital Earth, Chinese Academy of Sciences, Beijing 100101, China
Correspondence to:
O. Dubovik (**oleg.dubovik@univ-lille.fr**);
Y. Derimian (**yevgeny.derimian@univ-lille.fr**)

## 30  Abstract

This study presents a novel methodology for remote monitoring of aerosol
composition over large spatial and temporal domains. The concept is realized within
the GRASP (Generalized Retrieval of Aerosol and Surface Properties) algorithm to
directly infer aerosol composition from the measured radiances. This approach is
different from the conventional methods that use post-processing of the retrieved
aerosol optical properties for aerosol typing. The proposed method assumes observed
aerosols as mixtures of particles composed of black carbon, brown carbon, absorbing
insoluble, non-absorbing insoluble embedded in a soluble host. The algorithm then
derives size distribution and the fractions of these components. The complex
refractive index of each component is fixed a priori and the complex refractive index
of mixture is computed using mixing rules. The approach is first tested with synthetic
data and the uncertainties are estimated. Then, it is applied to the real ground-based
AERONET and space-borne POLDER/PARASOL observations, known to be





sensitive to aerosol complex refractive index. The study presents a first attempt to
derive aerosol composition from satellites. The obtained aerosol optical
characteristics are highly consistent with the standard products (R of ~ 0.9 for aerosol
optical thickness). The approach also presented an ability to separate between aerosol
properties in fine and coarse size fractions, in case of POLDER/PARASOL and
AERONET. Examples of application to POLDER/PARASOL on the global scale are
presented. The obtained spatial and temporal patterns of the aerosol composition
agree well with our knowledge on aerosol sources and transport features. Finally,
limitations and perspectives are discussed.

## 1 Introduction

Information about atmospheric aerosol chemical composition has a great
importance for monitoring and understanding of various aspects of climate and
environment. This information can be obtained by laboratory analysis of sampled
aerosol. However, the in-situ measurements require considerable effort and represent
only small areas without providing results on wide spatial and temporal scale. It is
known that chemical transport models are able to represent chemical component
concentrations with wide spatial and temporal coverage, and this capability has been
developed rapidly in the past decade. However, the models can have uncertainties
because they are initialized by gridded emission inventories that presently have
substantial uncertainties. For example, the carbon emissions inventories can be
uncertain with a factor of two, and this uncertainty is carried forward to the model
output (Bond et al., 1998; Cooke et al., 1999; Streets et al., 2001).
Aerosol components are often divided into two categories: strongly light-absorbing
components and mainly scattering (non-absorbing) components. The radiative
impacts of aerosols at the top of the atmosphere can change from cooling for highly
scattering aerosols to warming for highly absorbing aerosols located above highly
reflective surfaces like snow or clouds (Haywood and Shine, 1995). There are two
kinds of absorbing aerosols that are commonly found in the atmosphere: absorbing
carbon and mineral dust that contains iron oxides (Sokolik and Toon, 1999).
Light-absorbing carbon is composed of soot carbon and brown carbon (Andreae
and Gelencsér, 2006). Light-absorbing carbon is produced by incomplete combustion,
and it is an important component of atmospheric aerosol. The complex refractive
index of light-absorbing carbon is dependent upon the type of the fuel and the
conditions of combustion. Based on the origin of the material, combustion conditions,
morphological characteristics, chemical composition and optical properties (Andreae
and Gelencsér, 2006; Schkolnik et al., 2007), the term black carbon (BC), associated
with elemental carbon or soot, is used to define the strongly light-absorbing carbon in
the atmosphere. Meanwhile, the term brown carbon (BrC) is used for strongly
wavelength-dependent light-absorbing carbon particles whose absorption is greater at
near-ultraviolet and blue wavelengths (Chen and Bond, 2010; Dinar et al., 2007;
Hoffer et al., 2006; Jacobson, 1999; Kanakidou et al., 2005; Kirchstetter et al., 2004;
Schnaiter et al., 2006; Sun et al., 2007).



Mineral dust particles can have a strong spectral signature, with strong absorption
at the UV and blue wavelengths when iron oxides are present. Hematite and goethite
are different forms of free iron, and they typically appear together (Arimoto et al.,
2002; Formenti et al., 2014; Lafon et al., 2006; Shi et al., 2012). The presence of iron
in mineral dust particles is known to be important for its biogeochemical and radiative
impacts (Jickells et al., 2005; Mahowald et al., 2005; Sokolik and Toon, 1999).
Although the regional distribution of the iron concentration is important for climate
studies, it is difficult to obtain since it requires in-situ aerosol sampling or simulation
of complex natural processes. In addition, mineral dust particles can be affected by
the presence of anthropogenic aerosol particles (e.g. carbonaceous particles produced
from biomass burning). Separating the absorption associated with light-absorbing
carbon from the absorption associated with mineral dust (especially iron oxides) is not
an evident task (Derimian et al., 2008) and determination of the relative proportions
of BC, BrC and iron oxides should consider differences in absorption spectral
dependence. For instance, Dubovik et al. (2002a) showed that the spectral absorption
of carbonaceous aerosol is distinct from that of mineral dust. Schuster et al. (2005)
inferred the BC column content from AERONET retrievals by assuming BC is the
source of all significant aerosol absorption in the AERONET retrievals. Koven and
Fung (2006) retrieved hematite concentration at dust sites based on the spectral
variability of the imaginary refractive index, while Arola et al. (2011) retrieved BrC
from AERONET retrievals. Wang et al. (2013) have added single-scatter albedo as an
additional constraint to the approach using refractive index (Arola et al., 2011;
Schuster et al., 2005) and made it feasible to distinguish BC, BrC and dust
simultaneously. Similarly, Li et al. (2015, 2013) investigate the microphysical, optical
and chemical properties of atmospheric aerosols by fitting the AERONET complex
refractive indices measured at Beijing and Kanpur. Recently, Schuster et al. (2016)
have used the AERONET size distributions and complex refractive indices to retrieve
the relative proportion of carbonaceous aerosols (BC and BrC) and free iron minerals
(hematite and goethite) in fine and coarse modes particles. Nevertheless, all above
remote sensing methods (Arola et al., 2011; Koven and Fung, 2006; Li et al., 2015,
2013; Schuster et al., 2016, 2009, 2005; Wang et al., 2013) that retrieve aerosol
composition relying on an intermediate retrieval of the refractive index (e.g. one
provided by the AERONET operational inversion). We also note that these retrievals
of aerosol composition are only conducted for ground-based remote sensing
measurements.
Global satellite observations of aerosol properties provide an opportunity to
validate and constrain the model simulations at large spatial and temporal scales
(Collins et al., 2001; Liu et al., 2005; Yu et al., 2006, 2004, 2003; Zhang et al.,
2008a). The integration of observations with model results can fill gaps in satellite
retrievals and constrain global distributions of aerosol properties to have good
agreement with ground-based measurements (Liu et al., 2005; Yu et al., 2006, 2003).
In this regard, inverse modeling can be used to reduce large aerosol simulation
uncertainties. For instance, several studies (Chen et al., 2018; Dubovik et al., 2008;
Henze et al., 2007) showed the ability to retrieve global aerosol sources with inverse



models that rely upon satellite observations. Therefore, the practice of satellite data
fusion into models provides a possibility of improving aerosol simulations of the pre-
and post-satellite eras. However, besides the knowledge of amounts (concentrations)
and locations of aerosol emissions, an accurate modeling of atmospheric aerosols and
their effects also requires information about particle composition. The lack of
comprehensive datasets providing multiple constraints for the key parameters
employed in the models has hindered the improvement of model simulation.
Specifically, improving the ability of aerosol composition estimation will require
enhancement of remote sensing capabilities to provide the aerosol composition
information on the global scale. The accuracy and specification of the aerosol
composition as retrieved from satellite observations should respond to the
requirements of the chemical transport models. At the same time, the information
content of remote sensing is limited and the main challenge is to identify the aerosol
composition parameters that can be successfully retrieved by remote sensing
measurements, given their sensitivity to the aerosol optical properties and complex
refractive index in particular.
The POLDER space instrument (Deschamps et al., 1994; Tanré et al., 2011) is an
example of the instrument providing satellite observations that are sensitive to aerosol
composition. The implementation of multi-wavelength, multi-angle and polarization
measurement capabilities has made it possible to derive particle properties (size,
shape and absorption; Dubovik et al., 2011; Waquet et al., 2013) that are essential for
characterizing and estimating aerosol composition. This study presents a methodology
for the direct retrieval of aerosol composition from such measurements. Our
methodology is stimulated by the Schuster et al. (2016, 2009, 2005) works on
deriving aerosol composition information from ground-based Sun/sky photometers of
the AERONET network. Here, the idea of the approach has evolved and expanded for
retrieving the aerosol composition from satellite remote sensing observations as well.
Namely, we have incorporated an aerosol composition module into the Generalized
Retrieval of Aerosol and Surface Properties (GRASP) algorithm (Dubovik et al.,
2014, 2011). It should be noted that GRASP is a versatile algorithm designed to
retrieve an extended set of atmospheric parameters from diverse remote sensing data,
including surface, airborne, and satellite observations. Here, we apply GRASP to both
ground- and space-based observations, with primary objective to develop an approach
for monitoring aerosol composition with extensive spatial and temporal coverage.
The objective of our GRASP/Composition approach is to retrieve the aerosol
composition directly from remote sensing measurements without intermediate
retrieval of the complex refractive index, as in previous studies (Arola et al., 2011;
Koven and Fung, 2006; Li et al., 2015, 2013; Schuster et al., 2016, 2009, 2005; Wang
et al., 2013). This new approach has a more direct link to the measured radiance field
than the "intermediate" approaches, and we therefore expect a reduction in the
retrieval uncertainties. The GRASP/Composition approach also suggests an additional
constraint on the refractive index spectral variability that is not employed in the
conventional retrieval algorithms. Specifically the spectral variability of aerosol
complex refractive index is constrained in the GRASP/Composition retrieval by the



spectral dependences of the aerosol species used in the algorithm. It is expected that
such constraints can improve the retrievals in various situations.

One of the principal difficulties, however, is the identification of an adequate
conversion model for linking refractive index to aerosol composition. An ideal
conversion model should cover the entire range of aerosol complex refractive indices
and also provide a unique connection between spectral refractive index and aerosol
composition. Therefore, our primary objective focuses on identifying the optimal
transformation of chemical and physical aerosol information to optical properties (e.g.
refractive index). Once developed, the efficiency of the concept is verified and
demonstrated by applying GRASP/Composition to ground-based Sun/sky photometric
measurements, since this type of measurement usually presents a higher sensitivity to
aerosol absorption than the satellite remote sensing. Finally, the outcome of the
GRASP/Composition approach is demonstrated with the application of the aerosol
composition retrieval to multi-angular polarimetric POLDER/PARASOL satellite
observations.

## 191    2 Methodology

GRASP is a highly rigorous and versatile aerosol and surface reflectance retrieval
algorithm that is accessible at https://www.grasp-open.com (Dubovik et al., 2014,
2011). The essence of methodological developments in this study is to integrate a new
conversion model designed to link aerosol composition with optical and
microphysical characteristics into the standard GRASP inversion procedure. The
general logistics is shown in Fig. 1 (modified from (Dubovik et al., 2011)). The
algorithm is divided into several interacting but rather independent modules to
enhance its flexibility. The straightforward exchange of limited parameters minimizes
the interactions between the modules. The "Forward Model" and "Numerical
Inversion" are the two most complex and elaborate modules in the algorithm. The
"Forward Model" is developed in a quite universal way to quantitatively simulate the
measured atmospheric radiation with given surface and aerosol properties. The
"Numerical Inversion" module (which can be used in various applications, some not
even related to atmospheric remote sensing) includes general mathematical operations
unrelated to the particular physical nature of the observations. Numerical inversion is
implemented as a statistically optimized fitting of observations based upon the multi-
term least squares method (LSM), and combines the advantages of a variety of
approaches. The module provides transparency and flexibility for developing
algorithms that invert passive or active observations to derive several groups of
unknown parameters (Dubovik, 2004).

As a consequence of such organization of the algorithm, it can equally be applied
(with minimal changes) to invert observations from different satellite sensors or
ground-based instruments (Benavent-oltra et al., 2017; Espinosa et al., 2017; Lopatin
et al., 2013; Román et al., 2018, 2017; Tsekeri et al., 2017). A full description of the
"Forward Model" and "Numerical Inversion" algorithm modules can be found in
Dubovik et al. (2011). The following sections provide a description of the



modifications conducted for realization of the GRASP/Composition approach
(schematically presented by red dashed frames in Fig. 1).

## 2.1 Forward model


The formulation of the forward radiative transfer modeling in the presented
approach is generally similar to the formulation of the standard GRASP algorithm
where the modeling of the aerosol scattering matrices has been implemented
following the ideas described in Dubovik and King (2000) and Dubovik et al. (2006,
2002b). However, we implemented some modifications in modeling of aerosol single
scattering. Namely, the real and imaginary parts of the aerosol complex refractive
index are calculated using fractions of aerosol composition elements and fixed
refractive index of these elements as assumed in the conversion model. Thus, the new
composition approach uses the same forward model as described in Dubovik et al.
(2011), except that aerosol component fractions are iterated in the vector of the
retrieved unknowns (instead of refractive index) and refractive index is computed a
posteriori.
It is worth noting that the aerosol properties in the GRASP algorithm are retrieved
simultaneously with the surface reflectance characteristics. The land surface
Bidirectional Reflectance Distribution Function (BRDF) in GRASP is described by
the kernel-driven Ross-Li model. This model uses a linear combination of three
kernels $f_{iso}$, $f_{vol}$, and $f_{geom}$ representing isotropic, volumetric, and geometric optics
surface scattering, respectively (Li and Strahler, 1992; Roujean et al., 1992; Wanner
et al., 1995). The semi-empirical equation by Maignan et al. (2009) is used for the
Bidirectional Polarization Distribution Function (BPDF). The reflective properties of
ocean surface are modeled analogously to earlier POLDER algorithm developments
(Deuzé et al., 2001; Herman et al., 2005; Tanré et al., 2011). Fresnel reflection of the
agitated sea surface is taken into account using the Cox and Munk model (Cox and
Munk, 1954). The water leaving radiance is nearly isotropic (Voss et al., 2007) and
modeling shows that its polarization is negligible (Chami et al., 2001; Chowdhary et
al., 2006; Ota et al., 2010). The Fresnel term and the white cap reflection are taken
into account by Lambertian unpolarized reflectance. The whitecap reflectance is
driven by the wind speed at the sea surface according to the Koepke model (Koepke,
1984). The seawater reflectance at short wavelengths depends on the properties of
oceanic water and can be significant. Thus, in present model, the wind speed and the
magnitude of seawater reflectance at each wavelength are retrieved simultaneously
with the atmospheric aerosol properties.
The aerosol and surface characteristics are determined by parameters included in
the vector of unknowns and correspondingly they are inferred from observations.
Table 1 shows the list of measurements and retrieved parameters from
POLDER/PARASOL observations. For AERONET retrieval the list of parameters is
not shown here. However, in principle, it is analogous to POLDER/PARASOL, with





the difference that the set of observations is different (i.e. AERONET uses AOT and
transmitted total radiances at different wavelengths) and that surface parameters are
not retrieved but fixed from the climatology.

**2.2 Numerical inversion**

The numerical inversion implemented in this study follows the methodology
described in the paper of Dubovik et al. (2011). The only difference is that the
GRASP/Composition approach retrieves the fractions of different aerosol components
instead of the spectral dependence of the complex refractive index. Therefore, this
section describes only the modifications that are needed to implement the
GRASP/Composition approach.
GRASP retrieval is designed as a statistically optimized fitting routine and uses
multiple a priori constraints. GRASP can implement two different scenarios of
satellite retrievals: (i) conventional single-pixel retrieval for processing of satellite
images pixel by pixel and (ii) multiple-pixel retrieval for inverting a large group of
pixels simultaneously. The multi-pixel approach can be used for
POLDER/PARASOL data for improving consistency of temporal and spatial
variability of retrieved characteristic. The main modifications required for the
composition approach are related to the definition of a priori constraints.
Correspondingly, two types of a priori constraints are reformulated in the composition
retrieval approach: constraints for single pixel and constraints limiting inter-pixel
variability of derived parameters.
**2.2.1 Single-pixel observation fitting**

For each i-th pixel, the retrieval follows a multi-term LSM fitting of joint sets of
data combining the observations with a priori constraints defined by the system of
equations $\boldsymbol{f}_i^* = \boldsymbol{f}_i(\boldsymbol{a}_i) + \Delta\boldsymbol{f}_i$:

$$
\begin{cases}
\boldsymbol{f}_i^* = \boldsymbol{f}_i(\boldsymbol{a}) + \Delta\boldsymbol{f}_i \\
\boldsymbol{0}_i^* = \boldsymbol{S}_i\boldsymbol{a}_i + \Delta(\Delta\boldsymbol{a}_i) \Rightarrow \boldsymbol{f}_i^* = \boldsymbol{f}_i(\boldsymbol{a}_i) + \Delta\boldsymbol{f}_i \\
\boldsymbol{a}_i^* = \boldsymbol{a}_i + \Delta\boldsymbol{a}_i^*
\end{cases}
\qquad (1)
$$


Here, $\boldsymbol{f}_i^*$ denotes a vector of the measurements, $\Delta\boldsymbol{f}_i^*$ denotes a vector of
measurement uncertainties, $\boldsymbol{a}_i$ denotes a vector of unknowns in i-th pixel. The second
expression in Eq. (1) characterizes the a priori smoothness assumptions that constrain
the variability of the size distributions and the spectral dependencies of the retrieved
surface reflectance parameters. The matrix S includes the coefficients for calculating
the m-th differences of $dV(r_j)/dlnr$, $Frac(i)$, $f_{iso}(\lambda_i)$, $f_{vol}(\lambda_i)$, and $f_{geom}(\lambda_i)$. The
m-th differences are numerical equivalents of the m-th derivatives. $\boldsymbol{0}_i^*$ represents
vector of zeros and $\Delta(\Delta\boldsymbol{a})$ represents vector of the uncertainties that characterizes the
deviations of the differences from the zeros. This equation indicates that all of these
m-th differences are equal to zeros within the uncertainties $\Delta(\Delta\boldsymbol{a}_i)$. The third




expression in Eq. (1) includes the vector of a priori estimates $\boldsymbol{a}_i^*$, as well as the vector
of the uncertainties ($\Delta\boldsymbol{a}_i^*$) in a priori estimates of the i-th pixel.
The statistically optimized solution of Eq. (1) corresponds to the minimum of the
following quadratic form (according to multi-term LSM):

$$\Psi_i(\boldsymbol{a}_i) = \Psi_f(\boldsymbol{a}_i) + \Psi_\Delta(\boldsymbol{a}_i) + \Psi_a(\boldsymbol{a}_i)$$


$$= \tfrac{1}{2}((\Delta\boldsymbol{f}^P)^T(\boldsymbol{W}_f)^{-1}\Delta\boldsymbol{f}^P + \gamma_\Delta(\boldsymbol{a}_i)^T\Omega\boldsymbol{a}_i + \gamma_a(\boldsymbol{a}_i - \boldsymbol{a}_i^*)^T\boldsymbol{W}_a^{-1}(\boldsymbol{a}_i - \boldsymbol{a}_i^*)). \quad (2)$$

Following Dubovik et al. (2011), all equations are expressed with weighting
matrices W that are defined as $\boldsymbol{W} = (1/\varepsilon^2)\boldsymbol{C}$ (dividing the corresponding covariance
matrix C by its first diagonal element $\varepsilon^2$); the Lagrange multipliers $\gamma_a$ and $\gamma_\Delta$ are
written as $\gamma_\Delta = \varepsilon_f^2/\varepsilon_\Delta^2$ and $\gamma_a = \varepsilon_f^2/\varepsilon_a^2$, where $\varepsilon_f^2, \varepsilon_\Delta^2$, and $\varepsilon_a^2$ represent the first
diagonal elements of corresponding covariance matrices $\boldsymbol{C}_f$, $\boldsymbol{C}_\Delta$, and $\boldsymbol{C}_a$. Thus, in this
general formulation the component fractions ($Frac(i)$) of aerosol composition are
presented as unknowns instead of $n(\lambda_j)$ and $k(\lambda_j)$.

**2.2.2 Multiple-pixel observation fitting**
In this retrieval regime the fitting for a group of pixels is constrained by the extra a
priori limitations on inter-pixel variability of aerosol and/or surface reflectance
properties. Since the information content of the reflected radiation from a single pixel
is sometimes insufficient for a unique retrieval of all unknown parameters, the
presented approach can improve the stability of satellite data inversions (Dubovik et
al., 2011). The inversion of the multi-pixel observations is a solution for a combined
system of equations. For example, a three-pixel system can be defined as following:

$$\begin{cases} \boldsymbol{f}_1^* = \boldsymbol{f}_1(\boldsymbol{a}_1) + \Delta\boldsymbol{f}_1 \\ \boldsymbol{f}_2^* = \boldsymbol{f}_2(\boldsymbol{a}_2) + \Delta\boldsymbol{f}_2 \\ \boldsymbol{f}_3^* = \boldsymbol{f}_3(\boldsymbol{a}_3) + \Delta\boldsymbol{f}_3 \\ \quad\quad \dots \\ \boldsymbol{0}_x^* = \boldsymbol{S}_x\boldsymbol{a} + \Delta(\Delta_x\boldsymbol{a}) \\ \boldsymbol{0}_y^* = \boldsymbol{S}_y\boldsymbol{a} + \Delta(\Delta_y\boldsymbol{a}) \\ \boldsymbol{0}_t^* = \boldsymbol{S}_t\boldsymbol{a} + \Delta(\Delta_t\boldsymbol{a}) \end{cases}, \quad (3)$$

where the subscript "i" (i=1, 2, 3, …) is the pixel index. The total vector of unknowns
a is combined by the vectors of unknowns $\boldsymbol{a}_i$ of each i-th pixel, i.e.
$\boldsymbol{a}^T = (\boldsymbol{a}_1; \boldsymbol{a}_2; \boldsymbol{a}_3)^T$. The matrices $\boldsymbol{S}_x$, $\boldsymbol{S}_y$ and $\boldsymbol{S}_t$ include the coefficients for
calculating the m-th differences of spatial or temporal inter-pixel variability for each
retrieved parameter $a_k$ that characterizes $dV(r_j)/dlnr$, $Frac(i)$, $f_{iso}(\lambda_i)$, $f_{vol}(\lambda_i)$,
and $f_{geom}(\lambda_i)$. The vectors $0_x^*, 0_y^*, 0_t^*$ denote vectors of zeros and the vectors
$\Delta(\Delta_x\boldsymbol{a})$, $\Delta(\Delta_y\boldsymbol{a})$ and $\Delta(\Delta_t\boldsymbol{a})$ denote vectors of the uncertainties characterizing the
deviations of the differences from the zeros.




The statistically optimized multi-term LSM solution corresponds to the minimum
of the following quadratic $\Psi(\boldsymbol{a}^P)$:
$$\Psi(\boldsymbol{a}^P) = \left(\sum_{i=1}^{N_{pixels}} \Psi_i(\boldsymbol{a}^P)\right) + \frac{1}{2}(\boldsymbol{a}^P)^T \boldsymbol{\Omega}_{inter} \boldsymbol{a}^P. \qquad (4)$$

This is the sum of the corresponding single-pixel forms (first term) and an inter-pixel
smoothing component ($2^{nd}$ term). The smoothness matrix $\boldsymbol{\Omega}_{inter}$ in the inter-pixel
smoothing term is defined as:
$$\boldsymbol{\Omega}_{inter} = \gamma_x \boldsymbol{S}_x^T \boldsymbol{S}_x + \gamma_y \boldsymbol{S}_y^T \boldsymbol{S}_y + \gamma_t \boldsymbol{S}_t^T \boldsymbol{S}_t. \qquad (5)$$

Hence, the solution of a multi-pixel system of N pixels is not equivalent to the
solution of N independent single pixel systems.

### 2.2.3 A priori smoothness constraints of fitting

For the framework of deriving aerosol composition from the POLDER/GRASP
retrieval, the vector $\boldsymbol{a}_i$ is composed as:
$$\boldsymbol{a} = (\boldsymbol{a}_v \boldsymbol{a}_{frac} \boldsymbol{a}_{sph} \boldsymbol{a}_{Vc} \boldsymbol{a}_h \boldsymbol{a}_{brdf,1} \boldsymbol{a}_{brdf,2} \boldsymbol{a}_{brdf,3} \boldsymbol{a}_{bpdf})^T, \qquad (6)$$

where $\boldsymbol{a}_v$, $\boldsymbol{a}_{frac}$, and $\boldsymbol{a}_{sph}$ represent the constituents of the vector a corresponding to
$dV(r_i)/dlnr$, $Frac(i)$ and $C_{sph}$. Then $\boldsymbol{a}_h$ characterizes the mean altitude of the
aerosol layer $h_a$, the element $\boldsymbol{a}_{Vc}$ represents the total volume concentration, and $\boldsymbol{a}_v$
are the logarithms of $dV(r)/dlnr$ which are normalized by total volume
concentration. The three components ($\boldsymbol{a}_{brdf,1}, \boldsymbol{a}_{brdf,2}, \boldsymbol{a}_{brdf,3}$) are related to the
logarithms of the spectrally dependent parameters $k_{iso}(\lambda_i)$, $k_{vol}(\lambda_i)$ and $k_{geom}(\lambda_i)$
employed in Ross-Li model. The vector $\boldsymbol{a}_{bpdf}$ includes the parameters of the BPDF
model. Thus, this work differentiates from Dubovik et al. (2011) by retrieving the
volume fractions of the aerosol components (i.e. $Frac(i)$) instead of the complex
refractive index.
There is no evident connection between the retrieved fractions of aerosol
composition in each single pixel, so no smoothness constraints are used for $\boldsymbol{a}_{frac}$. The
matrix S for each i-th pixel is the same and has the following array structure (Dubovik
et al., 2011):



$$\boldsymbol{Sa} = \begin{pmatrix} S_v & 0000 & 0 & 0 & 0 & 0 \\ 0 & 0000 & 0 & 0 & 0 & 0 \\ 0 & 0000 & 0 & 0 & 0 & 0 \\ 0 & 0000 & 0 & 0 & 0 & 0 \\ 0 & 0000 & 0 & 0 & 0 & 0 \\ 0 & 0000 & S_{brdf,1} & 0 & 0 & 0 \\ 0 & 0000 & 0 & S_{brdf,2} & 0 & 0 \\ 0 & 0000 & 0 & 0 & S_{brdf,3} & 0 \\ 0 & 0000 & 0 & 0 & 0 & S_{bpdf} \end{pmatrix} \begin{pmatrix} \boldsymbol{a}_v \\ a_{frac} \\ a_{sph} \\ a_{Vc} \\ a_h \\ \boldsymbol{a}_{brdf,1} \\ \boldsymbol{a}_{brdf,2} \\ \boldsymbol{a}_{brdf,3} \\ \boldsymbol{a}_{bpdf} \end{pmatrix},$$ (7)


where the corresponding matrices $S_{...}$ have different dimensions and describe
differences of different order. The vectors in Eq. (7) corresponding to $\boldsymbol{a}_{frac}$, $\boldsymbol{a}_{sph}$,
$\boldsymbol{a}_{Vc}$, $\boldsymbol{a}_h$ contain only zeros because no smoothness constraint can be applied to these
parameters. The errors $\Delta(\Delta\boldsymbol{a})$ are assumed independent for different components of
the vector $(\Delta\boldsymbol{a})^*$ and the smoothness matrix for each i-th pixel can be written as:

$$\gamma_\Delta\Omega = \begin{pmatrix} \gamma_{\Delta,1}\Omega_1 & 0000 & 0 & 0 & 0 & 0 \\ 0 & 0000 & 0 & 0 & 0 & 0 \\ 0 & 0000 & 0 & 0 & 0 & 0 \\ 0 & 0000 & 0 & 0 & 0 & 0 \\ 0 & 0000 & 0 & 0 & 0 & 0 \\ 0 & 0000 & \gamma_{\Delta,2}\Omega_2 & 0 & 0 & 0 \\ 0 & 0000 & 0 & \gamma_{\Delta,3}\Omega_3 & 0 & 0 \\ 0 & 0000 & 0 & 0 & \gamma_{\Delta,4}\Omega_4 & 0 \\ 0 & 0000 & 0 & 0 & 0 & \gamma_{\Delta,5}\Omega_5 \end{pmatrix},$$ (8)


where $\boldsymbol{\Omega}_i = \boldsymbol{S}_i^T \boldsymbol{W}_i^{-1} \boldsymbol{S}_i$ uses the derivative matrices $\boldsymbol{S}_i$ (i=1, …, 5), $\boldsymbol{S}_v$, $\boldsymbol{S}_{brdf,1}$, $\boldsymbol{S}_{brdf,2}$,
$\boldsymbol{S}_{brdf,3}$, $\boldsymbol{S}_{bpdf}$.
The inter-pixel smoothing term given by Eq. (5) is defined in a very similar way as
described by Dubovik et al. (2011), and therefore it is not written here explicitly.
Indeed, the spatial and temporal variability of composition is very similar to the
variability of refractive index, since both depend only upon the variability of aerosol
type.
We note that the above equations apply to the POLDER/GRASP retrievals, but
corresponding equations are trivially obtained for the AERONET/GRASP retrievals
by excluding parameters describing surface reflectance and aerosol height.

**2.3 Model of optical properties of aerosol composition**
**2.3.1 Definition and assumptions**
The aerosol refractive index required for the forward calculations (see Fig. 1) is
derived by assuming a mixing model and employing fractions of aerosol species;
therefore, the retrieval of aerosol composition requires the selection of a mixing rule.
In our work, we decided to use a simple and widely tested Maxwell-Garnett effective



medium approximation. Indeed, the choice of the mixing rule is of importance since it
can significantly affect the retrieval results. In order to get an idea about the influence
of the mixing rule choice, the retrievals were produced also using the volume
weighted mixing rule and the results are inter-compared.
The Maxwell-Garnett mixing rule has been extensively applied in many studies for
retrieval of aerosol composition from ground-based remote sensing measurements (Li
et al., 2015, 2013; Schuster et al., 2016, 2009, 2005; Wang et al., 2013). As Fig. 2
illustrates, the first step in the Maxwell-Garnett conversion model is the designation
of a "host" and calculation of the refractive index of the host. In general, the host can
be formed by water and soluble inorganic species (e.g. ammonium nitrite, ammonium
sulfate, sea salt). It is well known that inorganic salt particles are mostly hygroscopic
and deliquescence in humid air. The phase transition from a solid particle to a saline
droplet (host) usually occurs when the relative humidity reaches a specific value,
known as the deliquescence point, that is specific to the chemical composition of the
aerosol particle (Orr et al., 1958; Tang, 1976; Tang and Munkelwitz, 1993). The
refractive indices of hygroscopic aerosols change with the additional amount of water
that is absorbed in response to changing relative humidity. These changes in refractive
index, including also the changes in specific density, size and mass fraction, have
been accurately measured as functions of relative humidity (Tang, 1996; Tang and
Munkelwitz, 1994, 1991).
In the presented approach, the host is assumed to depend upon the properties and
proportions of ammonium nitrate and water (uncertainties due to selection of
ammonium nitrate are evaluated further on). The real refractive index (at the 0.6328
μm wavelength) for a host mixture of ammonium nitrate and water can be expressed
as
$$n = 1.33 + (1.22 \times 10^{-3})X + (8.997 \times 10^{-7})X^2 + (1.666 \times 10^{-8})X^3, \qquad (9)$$
where X is the weight percent of ammonium nitrate (Tang and Munkelwitz, 1991).
Refractive indices at other wavelengths are spectrally interpolated utilizing
measured data (Downing and Williams, 1975; Gosse et al., 1997; Hale and Querry,
1973; Kou et al., 1993; Palmer and Williams, 1974; Tang, 1996; Tang and
Munkelwitz, 1991). A detailed description and FORTRAN subroutines for calculating
the host complex refractive index is accessible at the website of GACP (Global
Aerosol Climatology Project, https://gacp.giss.nasa.gov/data_sets/).
Once the refractive index of the host is determined, the refractive index of the
mixture is computed using the Maxwell-Garnett equations. The Maxwell-Garnett
effective medium approximation allows computation of the average dielectric
function based upon the average electric fields and polarizations of a host matrix with
embedded inclusions, and can model insoluble particles suspended in a solution
(Bohren and Huffman, 1983; Lesins et al., 2002).
The dielectric functions of aerosols are not typically tabulated in the literature, so
they must be computed from the refractive index. Once the dielectric functions are
known for the host and its constituents, the Maxwell-Garnett dielectric function for a





mixture can be calculated. For example, for two types of inclusions in a host, the
dielectric function of the mixture can be expressed as (Schuster et al., 2005):

$$\varepsilon_{MG} = \varepsilon_m \left[ 1 + \frac{3(f_1 \frac{\varepsilon_1 - \varepsilon_m}{\varepsilon_1 + 2\varepsilon_m} + f_2 \frac{\varepsilon_2 - \varepsilon_m}{\varepsilon_2 + 2\varepsilon_m})}{1 - f_1 \frac{\varepsilon_1 - \varepsilon_m}{\varepsilon_1 + 2\varepsilon_m} - f_2 \frac{\varepsilon_2 - \varepsilon_m}{\varepsilon_2 + 2\varepsilon_m}} \right], \tag{10}$$


where $\epsilon_m$, $\epsilon_1$, and $\epsilon_2$ are the complex dielectric functions of the host matrix and
inclusions, and $f_1$, $f_2$ are the volume fractions of the inclusions. If we use the case of
$f_2 = 0$, the corresponding complex refractive index of the mixture can be obtained by
Eqs. $(11) - (12)$:

$$m_r = \sqrt{\frac{\sqrt{\varepsilon_r^2 + \varepsilon_i^2} + \varepsilon_r}{2}}, \tag{11}$$


$$m_i = \sqrt{\frac{\sqrt{\varepsilon_r^2 + \varepsilon_i^2} - \varepsilon_r}{2}}, \tag{12}$$


where $\varepsilon_r$ and $\varepsilon_i$ denote the real and imaginary components of the mixture dielectric
function, $\varepsilon_{MG}$.
The selected refractive indices of inclusions in the Maxwell-Garnett effective
medium approximation model in this study are shown in Fig. 3. Figure 3 also
illustrates the assumption on the size resolved aerosol composition presented as an
additional constraint. Our selection of aerosol elements and the size resolved
composition results from the examination of a series of sensitivity tests and stability
of the inversion results. The size resolved composition formulation was chosen
because a similarity in spectral signatures of some aerosol species induced a difficulty
of their distinguishing in the considered in this study observational configurations.
For instance, brown carbon (BrC) and iron oxides (hematite and goethite) have
similar tendency in spectral absorption; that is, increasing the imaginary refractive
index towards ultraviolet wavelengths (Chen and Cahan, 1981; Chen and Bond, 2010;
Kerker et al., 1979; Schuster et al., 2016). At the same time, it is known that
carbonaceous absorbing aerosol particles dominate in the fine mode and mineral dust
absorption dominates in the coarse mode. Hence, black carbon (BC) and brown
carbon (BrC) are assumed to be the only absorbing insolubles in the fine mode and
iron oxides are assumed to be the only absorbing insolubles in the coarse mode. In
addition, the fine mode includes non-absorbing insoluble species (FNAI) that
represent fine dust or non-absorbing organic carbon (OC), non-absorbing soluble
species (FNAS) representing anthropogenic salts and aerosol water content (FAWC).
The coarse mode includes absorbing insoluble species (CAI), which are mainly iron
oxides, but can also include all other absorbing elements. The coarse mode also
includes non-absorbing insoluble (CNAI) species that mainly represent the bulk dust
material, but can be also non-absorbing insoluble organic carbon particles, non-



absorbing soluble species (CNAS) representing anthropogenic or natural salts (e.g.
sea salts) and aerosol water content (CAWC). It should be clarified that refractive
index of only one element is used for each species; however, our tests confirmed that
some elements are indistinguishable from the optical point of view, at least for the
measurement configurations expected in the scope of the presented algorithm
applications. Thus, several of the assumed species in the mixing model elements can
be associated with different elements.

**2.3.2 Sensitivity tests**
Using the above modifications to the GRASP algorithm described in Dubovik et
al. (2011), the aerosol composition retrieval approach was tested for inversion of
ground-based AERONET and POLDER/PARASOL satellite observations. For
verification of the proposed concept and the algorithm performance, a series of
sensitivity tests were conducted using synthetic data. A comprehensive series of
sensitivity tests were mainly conducted with the POLDER/PARASOL observations
because, unlike the AERONET retrievals, sensitivity of POLDER/PARASOL
observations to aerosol complex refractive index has not been systematically
explored. Thus, first, the POLDER/PARASOL radiances and polarization
measurements were simulated using forward calculations. Then, the synthetic
measurements were inverted using the GRASP algorithm with the size-dependent
aerosol composition approach and the Maxwell-Garnett mixing model. The tests were
conducted for a range of aerosol composition fractions for the species described
above and a variety of observational configurations such as spectral channels, viewing
geometry etc. Figure 4 presents an example of the assumed and retrieved fractions of
aerosol species in fine and coarse modes. The statistics of the sensitivity test results
are presented in Table 2, where we compare assumed and retrieved aerosol
parameters (fractions of aerosol elements, aerosol optical thickness (AOT), Single
Scattering Albedo (SSA) and complex refractive index at 675 nm). The results for
other wavelengths are very similar to that presented at 675 nm. In all the conducted
tests, the results demonstrated that in frame of the designed model the use of the size-
dependent Maxwell-Garnett conversion model allows the algorithm to distinguish
amongst the assumed aerosol species, including ammonium nitrate and water in the
host.

**2.3.3 Uncertainty assessment**
An important range of variability exists in the literature-reported refractive indices
of the aerosol species. Different assumptions on the refractive index of an aerosol
species can result in different retrieved fractions of the species proposed in this study.
To evaluate a possible range of the retrieved fractions due to uncertain knowledge of
the refractive indexes and difficulty to select one representative value, a series of
supplementary calculations were conducted using a range of refractive indices found





in the literature. Figure 5 shows the refractive indices employed in the algorithm and
those used for the assessment of the uncertainties in the retrieved aerosol fractions.
The tests are conducted as follows: first, synthetic measurements are created by
forward calculations while employing the complex refractive index assumed in the
algorithm; second, another complex refractive index is used in the inversion
procedure while retrieving the fractions of the aerosol species from the synthetic
measurements. Thus, the comparison of the assumed in the forward calculations and
the retrieved in the inversion procedure aerosol species fractions provides an error
assessment due to possible variability of their complex refractive index. The
calculations were conducted for all aerosol species that are assumed be embedded in
the host of the Maxwell-Garnett effective medium approximation. In addition, the
tests are also conducted for different fractions of the elements and for different values
of AOT, reflecting sensitivity of the retrievals to varying aerosol loading.

An extensive review of BC refractive indices can be found in Bond and Bergstrom
(2006) where the recommended imaginary part is in range from 0.63 to 0.79 at visible
wavelengths. The spectrally invariant value of 0.79 was adopted in the previous
studies (Bond et al., 2013; Bond and Bergstrom, 2006). Based on this literature, we
use the spectrally invariant complex refractive index for BC of $1.95 + 0.79i$ for our
current aerosol composition retrievals. We estimate then the uncertainty in the
retrieved BC fraction using a BC refractive index of $1.75 + 0.63i$. The results of the
uncertainty test for retrieving BC from POLDER/PARASOL are presented in Fig. 6a.
As can be seen, the uncertainty strongly depends on the BC fraction and increases
when the BC fractions are low. We note that the uncertainty can be large (over 100
%) when the BC fraction is below 0.01 and aerosol loading is weak. However, the
uncertainty decreases rapidly and can be 50 % or better for moderate and high aerosol
loading (AOT at 440 nm equal or more than 0.4) and when the BC fraction is above
0.01. Therefore, the estimates should be quite reasonable in cases of elevated
pollution.

The reported in the literature refractive index of BrC is variable. For the forward
model we employed the BrC refractive index derived from Sun et al. (2007), which
was used to retrieve aerosol composition from ground-based remote sensing
measurements (e.g. Arola et al., 2011; Schuster et al., 2016). The BrC refractive
index, representing carbonaceous particles with light absorption in the blue and
ultraviolet spectral regions emitted from biomass combustion (Kirchstetter et al.,
2004), was used for the uncertainty estimate. The tests show (Fig. 6b) that the
uncertainty in BrC fraction is more than 100 % when the fractions are below 0.1 and
decreases to below 100 % when the BrC fractions are above 0.1 and the aerosol
loading is elevated. Note that the uncertainty in BrC fraction is within 50 % when the
fractions are above 0.1 even for very low aerosol loading (AOT = 0.05).

Hematite and goethite are the dominant absorbers in the coarse mode particles. The
hematite refractive index was selected for the employed aerosol composition mixing
model. The literature shows that the hematite refractive indices can also exhibit quite
a large range of variability (e.g. see Fig. 5b). Figure 6c thus shows the uncertainties in
the retrieved CAI fraction from POLDER/PARASOL associated with the hematite



refractive given by Longtin et al. (1988) in the forward calculations and of Triaud
(2005) in the inversion. Except the very low fraction of CAI (below 0.005), the
uncertainty in CAI fraction is within 50 %.
The insoluble organic carbon and the non-absorbing dust present very similar
spectral dependence of complex refractive index (Ghosh, 1999; Koepke et al., 1997)
and it is practically impossible to distinguish between these species in the considered
in this work measurement configurations and the retrieval approach. Thus, the non-
absorbing insoluble organic carbon and non-absorbing mineral dust are expressed by
a non-absorbing insoluble species (NAI). The refractive index for the NAI in the
presented algorithm was taken as the dust refractive index in Ghosh (1999). The
uncertainty tests for the NAI fraction retrievals are presented for fine and coarse
fractions by replacing the dust refractive index with the refractive indices of dust
composed of quartz (Ghosh, 1999), kaolinite (Sokolik and Toon, 1999) and illite
(Sokolik and Toon, 1999) with the proportions of 48%, 26%, and 26%, respectively
(the proportions are recalculated from (Journet et al., 2014)) (see legend of Fig. 5).
The estimated uncertainties for fine and coarse NAI fractions (FNAI and CNAI)
decrease significantly (from 100 % to below 50 % and varying about the zero) when
the NAI fraction is above 0.1 (see Fig. 7a and 7b).
The non-absorbing insoluble can stand also for the insoluble organic carbon, as
was mentioned above. Thus, an additional test was conducted when the dust refractive
index (Ghosh, 1999) used in the forward calculations was replaced at the retrievals
stage by refractive index of insoluble organic carbon from Koepke et al. (1997). The
corresponding results of the retrieved in this case fine and coarse fractions of NAI for
POLDER/PARASOL observations are presented in Fig. 7c and 7d. The variability for
each fraction indicates that the choice of NAI refractive index can cause an
uncertainty in the retrieved NAI fraction less than 100% for FNAI and less than 50%
for CNAI when the fractions are above 0.1.
Figure 8 shows the uncertainties for the host species fraction (FNAS, CNAS,
FAWC, and CAWC), which are attributed to the differences between the refractive
indices and hygroscopic properties of ammonium nitrate and ammonium sulfate. The
uncertainties are small for FNAS, CNAS, FAWC, and CAWC, particularly when the
fractions are more than 0.2.

**3 Application to real remote sensing data**
**3.1 Composition retrieval from AERONET**

AERONET provides measurements that are among the most sensitive data to the
aerosol refractive index. In addition, the AOT in AERONET is result of direct
measurements and not retrieved as in the case of satellite observations. The GRASP
aerosol composition retrieval concept was therefore first tested with the real
AERONET data to check if the retrieved optical characteristics are consistent with the
results of standard AERONET product.



Figure 9 presents the AERONET measured AOT and Ångström Exponent (870
nm/440 nm) and retrieved Single-Scattering Albedo (SSA) at 675 nm versus those
retrieved using the GRASP/Composition approach. Namely, the operational
AERONET product is presented versus the derived from GRASP/Composition for 3
sites in the African continent: Banizoumbou (data for April 2007), Skukuza (data for
September 2007), and Ilorin (data for January 2007), representing according to the
sites location and the considered seasons the dust, the biomass burning and the
mixture of dust and biomass burning cases, respectively. It can be seen that the
aerosol optical properties are reproduced very well by GRASP/Composition approach
not only for the recalculated AOT and its spectral behavior, but also for the SSA. The
mean difference in AOT is about 0.01, which is on the level of the AERONET
calibration uncertainty, the difference in SSA is also well within the expected retrieval
uncertainty of 0.03 (Dubovik et al., 2002a). There is no biases observed and the
correlation coefficient is nearly 1.0 for AOT and Ångström Exponent.
It should be mentioned here that the fine mode in the presented retrievals is
described by 10 bins and the coarse mode by 15 bins, which is different than the 22
bins that are used for the entire size distribution in the standard AERONET algorithm
(Fig. 9). The composition retrieval has the ability to infer different refractive indices
for the fine and coarse modes. This is a significant improvement over the standard
AERONET and POLDER/PARASOL algorithms, which allow refractive indices to
vary with wavelength but not with size. Indeed, the use of fixed spectral dependences
of the refractive indices in the GRASP/Composition algorithm provides an additional
constraint and reduces the number of the unknown parameters. Thus, this approach
makes the inversion more stable. Nevertheless, the inter-comparison of retrievals by
the composition approach shows full consistency with the operational AERONET
product, mainly thanks to an additional physical constraint on the spectral dependence
of refractive index.
In addition to the better characterization of aerosol fine and coarse modes and
preserving consistency in retrievals of optical characteristics, the new approach can
also provide insights on aerosol composition. For example, Fig. 10 shows the volume
fractions of aerosol species retrieved in fine and coarse modes (panels a-f), and
fractions of the species in the total volume (panels g-i) for the mentioned above
African sites. The Banizoumbou site is located near Niamey (Niger) north of the
Sahel, the Ilorin site (Nigeria) is located in the Sahel, and the Skukuza site is located
in southern Africa. The retrieved aerosol compositions for Banizoumbou and Ilorin
present similarity in terms of abundant dust aerosol. However, contributions of BrC
and BC are strong in Ilorin, and the contribution of coarse absorbing insoluble aerosol
fraction is strong in Banizoumbou. The southern Africa site presents a different
picture: a strong contribution of coarse mode soluble and of fine mode non-absorbing
insoluble aerosol fractions attributed to water soluble organic carbon and water
insoluble organic carbon in the biomass burning region, and almost twice more
important than in the Sahel site contribution of BC. The BrC contribution, however, is
about two times smaller in southern Africa than that in Sahel, which is consistent with
AERONET's low spectral dependence for the imaginary index. The SSA in



Banizoumbou is highest (0.97 at 675 nm) and in Skukuza is lowest (0.82) because
dust and biomass burning aerosols dominate respectively in these two regions.

**3.2 POLDER/PARASOL satellite observations**

After testing the aerosol composition retrieval approach with the AERONET
measurements, the algorithm was applied to the POLDER/PARASOL satellite
observations. Figures 11, 12 and Table 3 summarize an inter-comparison of aerosol
optical characteristics derived by the GRASP/Composition approach applied for
POLDER/PARASOL and those of the operational AERONET product. The inter-
comparison is presented for six sites in Africa and Middle East (Fig. 11) and for all
available AERONET data (Fig. 12) representing performance for different aerosol
types and on the global scale. Because of a limited sensitivity to absorption when the
aerosol loading is low, the SSA product is filtered for AOT at 0.440 μm equal or
higher than 0.4 (Dubovik et al., 2002a; Dubovik and King, 2000). The SSA and the
Ångström Exponent in Fig. 11 are presented for all six sites together because the
dynamic range of the values for each single site is limited by a dominant aerosol type.
It should also be noted that in the inter-comparison on the global scale (Fig. 12) the
correlation for Ångström Exponent was notably better for higher AOT (R of ~ 0.6 for
all AOTs and of ~0.8 for AOT equal or more than 0.2). The better SSA and Ångström
Exponent retrievals for higher AOT is, however, known also for standard retrievals
and other satellite products (de Leeuw et al., 2015; Popp et al., 2016). Nevertheless,
the good agreements for AOT (R is generally of ~ 0.9 or better), and for Ångström
Exponent and SSA (R of ~ 0.70 - 0.80) show that the inversion of
POLDER/PARASOL satellite measurements using the composition approach is
consistent with the ground-based AERONET reference in terms of aerosol optical
properties. Analysis of the per site aerosol optical properties retrievals for different
aerosol types (Fig. 11) also does not reveal any evident problem.
The selected mixing model influence on the retrievals is assessed by comparison of
the results from Maxwell-Garnett effective medium approximation with performance
using a simplified volume-weighted (VW) aerosol mixture. Definition of the species
constituting the VW mixing model is quite similar to Maxwell-Garnett, it employs
BC, BrC in fine mode, absorbing insoluble in coarse mode, non-absorbing insoluble
and aerosol water content in both fine and coarse modes. The tests were conducted in
the same manner as for the Maxwell-Garnett effective medium approximation. The
sensitivity tests revealed that implementation of the volume-weighted mixing rule
yields stable results and this model can indeed be used for the retrievals. Moreover,
the VW model can be preferable in some applications due to its simplicity. Figure 11
and Table 3 illustrate that the GRASP/Composition retrievals using the MG and VW
mixing models almost equally well reproduce the aerosol optical properties. The
inter-comparison of the standard GRASP/PARASOL retrievals (without retrieval of
aerosol composition) with AERONET is also presented in Fig. 11 and Table 3. It





692 should noted that in all three shown cases the results obtained for AOT and Ångström
693 Exponent (AE) from PARASOL using the composition approach show comparable
694 and even better correlations with AERONET than the standard GRASP/PARASOL
695 retrieval that derives directly the spectral refractive indices instead of fractions of the
696 aerosol species with fixed refractive indices. This can be considered as confirmation
697 that the constraints adapted in the composition approach adequate provide realistic
698 and practically useful additional constraints that help to improve satellite retrievals.
699 At the same time, it can be seen that the SSA obtained by standard
700 GRASP/PARASOL correlates better with AERONET than those obtained by
701 GRASP/PARASOL composition approach. Specifically, this GRASP/PARASOL
702 composition shows systematically lower absorption than standard GRASP/PARASOL
703 retrieval. This can be explained by the fact that the complex refractive index in
704 GRASP/Composition are constrained by the information (on both magnitudes and
705 spectral tendencies) adapted from the literature while in case of standard
706 GRASP/PARASOL and operational AERONET products there are no such
707 constraints. As discussed by Dubovik and King (2000) and Dubovik et al. (2011) the
708 standard retrieval approach uses only smoothness constraints on spectral variability of
709 complex refractive index. In these regards, tests by Dubovik et al. (2000)
710 demonstrated that in presence of measurements noise the standard approach tends to
711 generate retrievals with higher values of absorption in the situation with lower aerosol
712 loading (lower AOD). This happens simply due to increased spread of SSAs for
713 situation with lower aerosol signal. Indeed, due to physical constraints SSA can not
714 higher than 1, as a result appearance of any spread caused by presence of the noise
715 generates lower SSA bias. Such bias has been often discussed by modelers
716 community as rather unfortunate feature of AERONET retrievals. Therefore, the
717 slightly higher SSA in case of GRASP/Composition can be considered rather a
718 positive effect of the additional constrain. Probably, additional focused analysis
719 should be done in future, but it can be expected that the slightly higher values in case
720 of GRASP/Composition may also on average be closer to what is expected from
721 models because tied to similar physical assumptions.

## 4 Illustration of global scale satellite aerosol composition retrieval

726 We processed the POLDER/PARASOL observations globally using the aerosol
727 composition retrieval algorithm. The results of this processing present the first
728 attempt to assess the measurement-based global distribution and seasonal variability
729 of aerosol composition. The data were processed for the year 2008, which provides a
730 notable variety of different aerosol types, including volcanic aerosols from a
731 Hawaiian eruption.

732 The results are further presented as seasonal means. It should be mentioned
733 however that any interpretation of the statistical values should take into account also



the number of available observations. Therefore, it is worth presenting the global
maps of the number of available cloud-free pixels. Figure 13 shows that the number
of the cloud free pixels over land is significantly higher than over ocean, which can
produce a difference in the mean values and create some artificial spatial patterns. In
addition, the sensitivity tests and experience of remote sensing observations treatment
show that the accuracy of the retrievals is low and the sensitivity to absorbing aerosol
and refractive index variability is particularly limited when the aerosol loading is low.
Therefore, it is also worth presenting the global maps of the aerosol optical thickness
(Fig. 14), prior to analyzing the aerosol composition retrievals. It should also be
outlined that despite the fractions of the elements are the initial retrieval parameters,
direct interpretation of the maps of these fractions can be confusing because do not
always correspond to a significant aerosol concentration. For instance, a large fraction
of an element retrieved for a size mode where aerosol volume concentration is very
low, can have no significant meaning as not having contribution to the optical signal.
Therefore, the volume concentrations of the retrieved elements and not the fractions
will be further presented. Figures 15 to 20 thus show seasonal variabilities of the
retrieved aerosol volume concentrations for different aerosol species.

## 4.1 Black Carbon

The retrieved aerosol composition shows patterns of biomass burning in the Sahel
and southern Africa regions, expressed by elevated concentrations of BC (Fig. 15).
The derived BC concentrations show a pronounced seasonal and spatial variability.
The largest concentrations can be observed over the African continent, another
noticeable region is Asia, namely India and China. The most intensive BC emissions
appear during DJF, which is constituted from contributions of the Sahel region, India
and China. Somewhat lower concentrations during SON and JJA are attributed to
biomass burning regions in southern African. A global minimum of the BC
concentrations is during MAM. The obtained spatial and seasonal patterns of BC are
consistent with the knowledge that DJF is the season of intense agricultural burning
across the sub-Sahelian region of Africa. BC generated from such agricultural burning
can extend for thousands of kilometers from east to west across the continent, as can
be seen in Fig. 15. The BC concentration in northern Africa appears mainly over land
near the west coast, especially from Senegal south to Gabon on the equator, and over
the Gulf of Guinea, which is attributed to the biomass burning during DJF (e.g.
Haywood et al., 2008). The BC observed over the ocean is generally transported from
biomass burning areas by prevailing trade winds. The retrievals show that the BC
concentration in India and China, which can be rather attributed to anthropogenic
activity, is maximal during DJF. This result is consistent with a previous study by Li
et al. (2015) that also found a maximal BC mass concentration during DJF. The work
by Li et al. (2015) is based on retrieval of aerosol composition from AERONET
measurements in Beijing and Kanpur sites and presents twelve years' climatology for
the period 2002 - 2013.





During JJA and SON, the elevated BC concentrations are mostly over southern
Africa, which is in line with the known African monsoon cycle. The variations of the
retrieved BC are consistent with the biomass burning activity progressing from north
to south Africa, starting from June, peaking in July - August and then decreasing in
intensity until late October with the end of the dry season (Cahoon et al., 1992;
Liousse et al., 1996; Maenhaut et al., 1996; Swap et al., 1996).
It should be reminded, however, that sensitivity to the absorption and therefore to
the BC signal is limited when the AOT is low. In addition, for very low AOT values
the aerosol volume concentrations are also low and therefore the retrieved fractions of
the aerosol species are more uncertain. Very low aerosol loading is typical for over
ocean observations (Fig. 14) and thus appearance of some BC concentrations over
ocean should be interpreted with caution.

**4.2 Brown Carbon**
Similar to BC, the observed patterns of BrC (Fig. 16) show seasonal variations,
primarily association with the biomass burning in Africa and the contribution of
Asian anthropogenic activities. A closer comparison of BrC and BC concentrations
reveals, however, that their maximal concentrations are not always collocated. This
observation reflects that fresh biomass burning aerosols have higher BC content than
aged aerosols (Abel et al., 2003; Haywood et al., 2003; Reid et al., 1998). During
SAFARI-2000, for example, the single scattering albedo has an increase from 0.84 to
0.90 between smoke close to the source and aged haze 5 h downwind from a large fire
(Abel et al., 2003), which is attributed to changes in aerosol composition. There can
be some rapid changes occurring in the relative concentration of particle types with
the aging of smoke and the BC particles become gradually more aggregated with
organic and sulfate particles during the aging of smoke (Pósfai et al., 2003).
Therefore, the more abundant presence of particles with the spectral absorption
signature of BC is reasonable for the areas near the biomass burning emissions,
whereas particles with a spectrally dependent absorption signature of BrC are
generally enriched in downwind region, which can explain appearance of BrC
concentrations in aerosol particles transported over ocean in northern hemisphere.

**4.3 Fine mode Non-Absorbing Soluble**
The NAS component is represented by the real part of the refractive index of
ammonium nitrate; however, sulfates, sea salt or aged hydroscopic particles are also
included in the NAS component. Figure 17 presents seasonal means of the NAS
retrieved for the fine mode (FNAS). The FNAS volume concentration dominates over
China and India, especially during DJF and SON, which can correspond to industrial
aerosol and heating activity in megacities with high population density. The spatial
patterns of FNAS also coincide with the patterns of BC in southern Africa that
indicates presence of non-absorbing particles fraction in the biomass burning



emissions (e.g. water soluble organic carbon). Indeed, the carbonaceous organic
particles can provide a favorable surface for aging processes and sulfate nucleation
(Li et al., 2003). Pronounced FNAS particles concentrations are retrieved during JJA
over the Mediterranean Sea region, which is in line with the knowledge on abundant
presence of anthropogenic and biogenic sulfate particles in the Mediterranean region
(Ganor et al., 2000; Lelieveld et al., 2002; Levin, 2005; Levin et al., 1996). The
FNAS particles are also retrieved south from the Mediterranean Sea, deep inland over
Libya and Egypt. This FNAS component can be possible in this area considering
persistent north-south, north-east air mass transport in the eastern Mediterranean
region governed by semi-permanent low-pressure trough extending in JJA from the
Persian Gulf (Bitan and Sa'Aroni, 1992).

## 4.4 Coarse mode Non-Absorbing Insoluble

In the northern and western Africa, the coarse mode non-absorbing insoluble
component appears all year long with the pronounced maximum concentrations
during MAM to JJA (Fig. 18), representing the non-absorbing part of mineral dust.
Notable is a shift in the maximum of this component towards higher latitudes in JJA
that corresponds to the northern shift of the inter-tropical convergence zone. The
retrievals also clearly show a "hot spot" of coarse mode non-absorbing dust over the
Bodélé depression, located between the Tibesti Mountains and Lake Chad, and known
as the most active dust source in the Sahara desert (Gasse, 2002; Prospero et al., 2002;
Washington et al., 2003). This dust source is caused by the coincidence of an
extensive source of diatomite sediment and high velocity winds associated with the
Bodélé Low Level Jet (Todd et al., 2007; Washington et al., 2006; Washington and
Todd, 2005) with the emission peaks during DJF and MAM (Herrmann et al., 1999;
Koren and Kaufman, 2004; Todd et al., 2007; Washington and Todd, 2005) that are
also distinguishable in the presented retrievals. This CNAI aerosol type also appears
over the Middle East, the Arabian Peninsula and extends over Asia, which is known
as the global dust belt. The coarse mode non-absorbing dust concentration is
particularly high over the Arabian Peninsula, central to southern Pakistan, as well as
over the Oman and Arabian seas. Over this region, the maximum dust concentration is
observed during MAM and JJA, while dust concentration substantially decreases
during SON and DJF. Higher dust concentration during MAM and JJA is primarily
caused by the strong northwesterly winds known as "Shamal Wind" and dry
conditions. The JJA peak is caused by several major sources of dust that have
maximum dust activity during JJA, including desert areas in Syria and Iraq where a
strong northwesterly Shamal Wind is blowing (Choobari et al., 2014). The Sistan
region can also be distinguished among the high dust concentrations. This region is
considered as a major dust source in southwest Asia (Ginoux et al., 2012; Goudie,
2014; Léon and Legrand, 2003; Middleton, 1986a) attributed to the strong persistent
northeasterly winds (Alizadeh Choobari et al., 2013; Middleton, 1986b; Miri et al.,
2007). This source can cause frequent dust and sand storms, especially during the
period of June to August contributing to the deterioration of air quality (Rashki et al.,



2013). In addition, during DJF and SON some elevated CNAI concentrations are
observed in Australia (area of Lake Eyre and The Great Artesian Basin). It should be
also noted that some CNAI concentrations are retrieved during the seasons and over
the regions in Africa known for biomass burning and over south-east of USA. These
concentrations indicate presence of some coarse mode non-absorbing particles
possibly of organic origin.

**4.5 Coarse mode Absorbing Insoluble**
The Coarse mode Absorbing Insoluble (CAI) particles, which mainly represent the
iron oxides contained in mineral dust, are generally associated with the desert regions
and with the elevated concentrations of CNAI. The high CAI concentrations are
observed during MAM and JJA over western Africa and the Arabian Peninsula (Fig.
19). High CAI concentrations are also retrieved over Asia during the same MAM and
JJA seasons and are quite clearly attributed to the region of the Taklimakan desert
located in northwest China. It is worth noting that the maximum of CAI and CNAI do
not always coincide, reflecting different percentage of iron oxides in desert dust that
is varying depending on the soil mineralogy of the source region. Calculations of the
ratio of CAI to CNAI concentrations over African continent provided values of up to
about 0.05, which is consistent with up to 3 to 5 % iron oxides in desert dust (e.g.
Ganor and Foner, 1996; Guieu et al., 2002; Zhang et al., 2003; Lafon et al., 2004).
The high CAI concentrations over western Africa are mainly present over Niger,
Mauritania and near the west coast. This is in line with a study by Formenti et al.
(2008) that demonstrates the higher iron oxide content in Sahelian dust originated
from the Sahel belt, while a lower content is in the Chad basin. Lázaro et al. (2008)
also reported that the iron oxide content of dust transported to the Canary Islands,
near the west coast, tends to have higher values for source areas between 0°N - 20°N.
In addition, high CAI concentrations are also derived over the Arabian Peninsula and
the Arabian Sea, which may be attributed to the dust originated from Saudi Arabia,
known for presence of an important iron content (Krueger et al., 2004). It should be
mentioned here that a discontinuity in the retrieved concentrations can be noted
between over land and over water in the regions of the Red Sea and Arabian Sea.
Given that such discontinuity does not appear in all coastal regions, but only in
particular circumstances, we suppose that there are some physical explanations. For
instance, the observed discontinuity corresponds well to the land topography, i.e.
presence of surrounding mountains and the observed in other work accumulation of
aerosol over the Red Sea (Brindley et al., 2015). It is also interesting to admit that
some coarse mode absorbing aerosol appear in the regions and seasons associated
with biomass burning and elevated concentrations of BrC and BC in the fine mode,
e.g. in Africa during DJF and SON seasons. This fact can reflect presence of
absorbing carbonaceous material in the coarse mode, which was fitted by refractive
index of iron oxide assumed as only the absorbing component of the coarse mode.



## 4.6 Fine mode Non-Absorbing Insoluble

Because the fine mode non-absorbing insoluble component (Fig. 20) can stand for
both OC and non-absorbing dust, the Ångström Exponent can be used as an additional
post retrieval criteria for a better interpretation of this component. For instance, the
joint FNAI and Ångström Exponent (maps are presented in supplementary material)
analysis shows that the particles concentrations derived over western Africa, Middle
East, Central Asia and northwest China mainly reflect presence of fine mode non-
absorbing dust because are associated with the values of Ångström Exponent
generally well below one. Specific examples are the concentrations derived over the
Bodélé depression during DJF, the Taklimakan desert in China during MAM and
Arabian Peninsula during JJA. However, the elevated FNAI particles concentrations
retrieved over southern Africa and South America during JJA and SON, over eastern
part of China and Siberia during JJA, and generally over India, are associated with
high values of Ångström Exponent, thus should rather be classified as organic carbon.
For example, high OC in southern China (Sichuan Basin and the Pearl River Delta
region) and urban south Asia is confirmed in several previous studies (Decesari et al.,
2010; Stone et al., 2010; Zhang et al., 2008b; Zhang et al., 2012). The OC of urban
origin in China is enhanced around May to June and October (Zhang et al., 2012),
which may be an explanation of the retrieved high OC concentration during JJA in
southern China. Secondary OC (SOC) can also contribute to the total concentrations
of OC (Miyazaki et al., 2006; Weber et al., 2007; Zhang et al., 2008b; Zhang et al.,
2005) and be retrieved here as FNAI. Additionally, the elevated OC concentration
over South America during SON correspond well to the known season of biomass
burning that starts in July and peaks generally in August and September (Duncan et
al., 2003).
A plume structure of elevated fine non-absorbing insoluble (Fig. 20) and soluble
(Fig. 17) components originated from Hawaiian Islands in the North Pacific Ocean is
also notable. This structure is visible during three seasons from MAM to SON and
corresponds to a Hawaiian volcano emission. The material emitted into the
atmosphere in this case was not the coarse volcanic ash, but continuous gaseous
emissions that can form secondary aerosol during downwind transport (Craddock and
Greeley, 2009; Edmonds et al., 2013). Identification of this material by the suggested
approach as a mixture of components equivalent to ammonium sulfate and fine non-
absorbing dust is therefore quite plausible.
## 4.7 Aerosol Water Content and Coarse mode Non-Absorbing Soluble

The algorithm also provides aerosol water content that is required to create the host
by mixture with non-absorbing soluble component. As result, the retrieved spatial and
temporal patterns of aerosol water content and non-absorbing soluble are very similar.
That is, the fine mode aerosol water content is mainly retrieved in the regions with



high loading of anthropogenic aerosol, similarly to the fine mode non-absorbing soluble. For instance, the fine mode aerosol water content can be seen over India and China during SON and DJF, at high latitudes of northern hemisphere and over Eurasia during SON. Some notable water concentrations are also retrieved over southern Africa during the biomass burning season (JJA), but mainly over ocean that correspond to visibly transported and likely aged aerosol. The maps of FAWC are presented in the supplementary material as they are very similar to already presented FNAS (Fig. 17).

The retrieved coarse mode aerosol water content and coarse non-absorbing soluble also present very similar spatial and temporal patterns. However, they are different from the patterns of fine mode. The concentrations are very low everywhere, except over ocean in the regions associated with high concentrations of the coarse non-absorbing insoluble (dust) component. This feature is associated with dust transported from western Africa and Arabian Peninsula. These coarse mode AWC and NAS retrievals require a careful interpretation. First, it should be realized that even relatively small aerosol water fraction retrieved in the regions of very high aerosol concentration can result in a pronounced volume concentration. In addition, aerosols with low real refractive index, which cannot be fully explained by the assumed dust aerosol model, will be interpreted as a water fraction. For instance, some low water aerosol concentration erroneously appears over the Bodélé depression during DJF. The Bodélé dust, however, is known to contain much fossil diatom (Formenti et al., 2008), which would have a different real part of refractive index then assumed in this study mixture of quartz, kaolinite and illite. At the same time, possible hygroscopicity of mineral dust, its coating by organics and internal mixture with sea salt, were found in several laboratory and field studies (e.g. Usher et al., 2003; Falkovich et al., 2004; Laskin et al., 2005; Derimian et al., 2017). The fact that the notable aerosol water content is observed in the retrievals only over ocean and not over land, except for retrievals over Bodélé, also agrees with hypothesis of the dust hygroscopicity. We therefore conclude that despite this pronounced water aerosol content in the coarse mode should be questioned and interpreted with caution, a physical significance of this result should not be excluded. Indeed, this retrieval result may not be fully understood at present but it was not enforced by any specific assumption or measurement artifact, and therefore it is likely to represent a manifestation of specific physical or chemical transformation of aerosol or properties of dust. In addition to the described above main feature of CAWC and CNAS, the derived maps are presented in supplements, together with the maps of FAWC, which are similar to already presented FNAS.

# 5 Conclusions

We present a new approach for monitoring atmospheric aerosol composition with remote sensing observations. Unlike existing aerosol composition retrieval algorithms that interpret an intermediate retrieval of the refractive index, this study utilizes a





direct fit of measurements. We demonstrate retrievals of several aerosol components
in fine and coarse size modes under assumption of an internal aerosol mixing rule.
The tests using a volume weighted mixing rule were also conducted and the results
compared.
The approach is implemented in a state of the art GRASP algorithm (Dubovik et
al., 2014, 2011) designed to process space-borne and ground-based remote sensing
observations. The composition module is incorporated in GRASP thus the new
GRASP/Composition version of the code employs mixtures of aerosol components
with known refractive indices. This approach serves also as an additional physical
constraint on spectral dependences of complex refractive index. The composition
module uses the Maxwell-Garnett effective medium approximation (EMA) and is
based on the Schuster et al. (2016, 2009) approach, but assumes independent aerosol
mixtures in the fine and coarse modes and the direct fit of radiances instead of an
intermedia step of fitting the retrieved refractive indices.
A series of numerical sensitivity tests with synthetic data were conducted to
evaluate the composition retrieval. Results of the tests showed that the new
conversion module allows the retrieval to distinguish amongst several assumed
aerosol components. The tests with the new module also show consistency with
GRASP tests that are traditionally configured for ground-based AERONET
measurements.
We also tested the algorithm with real measurements. Application of the
GRASP/Composition algorithm to the AERONET Sun/sky photometric
measurements retrievals of AOT, Ångström Exponent and SSA presented good
agreement with the standard operational AERONET product for sites dominated by
dust, biomass burning, and mixtures of dust and biomass burning aerosol. In addition,
because of the reduced number of parameters (instead of 8 parameters for complex
refractive index retrievals using 6 parameters for composition retrievals) and an
additional physical constrain on spectral dependence of refractive index in the
composition retrieval, the GRASP/Composition approach applied for AERONET can
split the characteristics of fine and coarse mode aerosol. The GRASP/Composition
algorithm was also applied for the POLDER/PARASOL satellite observations. An
inter-comparison of aerosol optical characteristics derived from POLDER/PARASOL
using the composition approach and those of the AERONET operational product
demonstrated a high reliability of the results.
The performance of the aerosol composition algorithm has been demonstrated by
the application to POLDER/PARASOL observations on the global scale for year
2008. The obtained spatial and temporal patterns of aerosol composition distribution
seem to agree well with known physical expectations. For a proper interpretation of
the obtained results it should be also realized that the retrieved aerosol species and
their concentrations compose a set of parameters that reproduces well the measured
radiation field and provides adequate retrieved optical properties of aerosol. At the
same time, the direct interpretation from the chemical point of view is not always



evident and even possible. For instance, as mentioned in the methodology part, the distinguishing of some chemical species is not possible for the given configuration of remote sensing measurements. However, the retrieved composition still reflects the aerosol microphysics and chemistry, and their variability. One should also remember that, based on the sensitivity tests and experience of aerosol characterization by remote sensing, the accuracy of the retrievals depends on the aerosol loading (AOT). Accuracy of the absorbing components retrieval can be primary affected. Thus, interpretation of all the obtained patters requires a more detailed analysis and it is realized that some erroneous composition features can be possible. The principle limitations of the presented approach are: (i) lack of sensitivity to absorption species in case of low AOT; (ii) difficulty to distinguish between iron oxide and absorbing carbonaceous species (BrC and BC), which is mainly related to the limited number of spectral channels in shortwave solar spectrum; (iii) non-absorbing insoluble component can include organic material, but also non-absorbing dust. These assumptions can lead to some misinterpretation; for instance, the analysis of the BrC retrievals at some locations reveals that the aerosol absorbing properties attributed to BrC should be attributed to the iron oxides that are present in the fine size fraction. A post-retrieval classification is helpful to resolve the shortcomings. For example, analysis of Ångström Exponent can indicate dominance of coarse particles of mineral dust origin or fine particles of combustion origin, which can provide more information about the non-absorbing insoluble component.

Nevertheless, the results are encouraging. For example, the derived BC and BrC exhibit a seasonal and spatial variability that is attributed to the known biomass burning season cycle in Africa and the anthropogenic pollution patterns in Asia, in particular India and China. Coarse mode absorbing (mainly iron oxides) and non-absorbing (mainly dust) insoluble components show a similar seasonal and spatial variability, reaching a peak during MAM and a minimum during SON. It is also noted that the maximums of iron oxide concentration are not co-located with those of dust, because the elemental and mineralogical compositions of mineral dust vary depending on the source region. The global dust belt extending from western Africa, through the Middle East to Central Asia is also observed in the composition retrieval. GRASP/Composition indicates high concentrations of non-absorbing insoluble appear over the Sahara, Arabian Peninsula, Caspian Sea and Aral Sea regions in Central Asia, and the Gobi and Taklimakan desert in China. In addition, dust was also detected over some regions in Australia during DJF and SON.

The composition retrieval algorithm demonstrated here using AERONET and POLDER/PARASOL data can also be used for interpreting other observations. That is, the composition approach is now incorporated in the GRASP algorithm, which has a generalized input and can be easily modified and adapted to other both passive and active remote sensing instruments, for example, the Directional Polarimetric Camera (DPC) launched onboard the GaoFen-5 Satellite in Chinese High-resolution Earth Observation Program, which is the first Chinese multi-angle polarized earth observation satellite sensor (Dubovik et al., 2019; Li et al., 2018). Moreover, the



proposed aerosol parameterization using components can be helpful not only for retrieving additional information about aerosol composition, but also for optimizing retrieval stability.

Additionally, we tested the volume-weighted mixing model, in addition to the Maxwell-Garnett EMA, to evaluate the sensitivity of our approach to the assumed aerosol EMA. We tested both approaches using our suite of aerosol species (i.e. BC, BrC, coarse mode absorbing insoluble, fine and coarse mode non-absorbing insoluble). The sensitivity tests revealed that implementation of the volume-weighted mixing rule also presents stable results that are consistent with the Maxwell-Garnett EMA. Thus, the volume-weighted model can also be employed in GRASP/composition retrieval, and may be preferable in some applications due to its simplicity.

The results of the aerosol composition retrieval from AERONET and POLDER/PARASOL satellite measurements demonstrate a potential for constraining global and regional aerosol modeling that can be particularly valuable because no other aerosol composition data are often available on a large spatial and temporal scale.

**Data availability**: The retrievals can be requested directly from the corresponding author (oleg.dubovik@univ-lille.fr or yevgeny.derimian@univ-lille.fr)

**Author contributions:** LL, OD, YD and GLS contributed to retrieval algorithm development and conducted the sensitivity and uncertainty tests using synthetic data. TL, PL, FD, DF and CC contributed to the modifications of GRASP code. ZL, AL, BT and HC contributed to the retrievals derived from AERONET measurements. LL, OD and YD wrote the paper with input from all authors.

**Competing interests**: The authors declare that they have no conflict of interest.

**Acknowledgments**

This work is supported by the Labex CaPPA (Laboratory of Excellence − Chemical and Physical Properties of the Atmosphere) project, which is funded by the French National Research Agency (ANR) through the PIA (Programme d'Investissement d'Avenir) under contract "ANR-11-LABX-0005-01". This work is also financially supported by grant from National Natural Science Foundation of China (41875157), National Key R&D Program of China (2016YFA0601901). The authors thank CNES and ICARE data distribution center for POLDER/PARASOL data and the entire AERONET team, especially the principal investigators of the AERONET sites used in this study, for their long-term efforts to maintain AERONET observation.



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



**Tables**
**Table 1**. List of measured and retrieved characteristic considered in POLDER/GRASP with
aerosol composition mixing model. $\mu_0 = cos(\vartheta_0)$ depends on the solar zenith angle $\vartheta_0$,
$\mu_1 = cos(\vartheta_1)$ depends on the observation zenith angle $\vartheta_1$. $\varphi_0$ and $\varphi_1$ represent the solar and
observation azimuth angles.

| POLDER/PARASOL measurements | |
|---|---|
| Measurements type: | |
| $I(\mu_0; \mu_1; \varphi_0; \varphi_1; \lambda_i) = I(\Theta_j; \lambda_i)$ | – I reflected total radiances |
| $Q(\mu_0; \mu_1; \varphi_0; \varphi_1; \lambda_i) = Q(\Theta_j; \lambda_i)$ | – Q component of the Stokes vector |
| $U(\mu_0; \mu_1; \varphi_0; \varphi_1; \lambda_i) = U(\Theta_j; \lambda_i)$ | – U component of the Stokes vector |
| Observation specifications: | |
| Angular: | |
| | $I(\Theta_j; \lambda_i), Q(\Theta_j; \lambda_i)$ and $U(\Theta_j; \lambda_i)$ measured in up to 16 viewing directions, that may cover the range of scattering angle $\Theta$ from ~ 80° to 180° |
| Spectral: | |
| | $I(\Theta_j; \lambda_i)$ measured in 6 window channels $\lambda_i = 0.440, 0.490, 0.565, 0.670, 0.865$ and $1.02\ \mu m$ |
| | $Q(\Theta_j; \lambda_i)$ and $U(\Theta_j; \lambda_i)$ measured in 3 window channels $\lambda_i = 0.490, 0.670,$ and $0.865\ \mu m$ |
| Retrieved characteristic | |
| Aerosol parameters: | |
| $C_v$ | – total volume concentration of aerosol ($\mu m^3/\mu m^2$) |
| $dV(r_i)/dlnr$ | – ($i = 1, ..., N_r$) values of volume size distribution in $N_i$ size bins $r_i$ normalized by $C_v$ |
| $C_{sph}$ | – fraction of spherical particles |
| $h_0$ | – mean height of aerosol layer |
| $Frac(F_i)$ | – ($i = 1, ..., N_f$) the fraction of composition in fine mode |
| $Frac(C_i)$ | – ($i = 1, ..., N_c$) the fraction of composition in coarse mode |
| Surface reflection parameters: | |
| Ross-Li model parameters: | |
| $k_{iso}(\lambda_i)$ | – ($i = 1, ..., N_\lambda = 6$) first Ross-Li model parameter (isotropic parameter characterizing isotropic surface reflectance) |
| $k_{vol}(\lambda_i)$ | – ($i = 1, ..., N_\lambda = 6$) second Ross-Li model parameter (volumetric parameter characterizing anisotropy of reflectance) |
| $k_{geom}(\lambda_i)$ | – ($i = 1, ..., N_\lambda = 6$) third Ross-Li model parameter (geometric parameter characterizing anisotropy of reflectance) |
| Maignan et al. (2009) model: | |
| $B(\lambda_i)$ | – ($i = 1, ..., N_\lambda = 6$) free parameter |






**Table 2**. List of statistics for parameters between assumed and retrieved in the sensitivity
tests of GRASP composition retrieval using Maxwell-Garnett mixing model. The values of
slope (A), intercept (B), correlation coefficient (R), root-mean-square error (RMSE), mean
absolute error (MEA), mean relative error (MRE) and standard error deviation (STD) are
presented for aerosol compositions, aerosol optical thickness (AOT), Single-scattering albedo
(SSA), real ($n$) and imaginary ($k$) parts of complex refractive index in fine mode (FM) and
coarse mode (CM) at 675 nm.

|         | A    | B     | R    | RMSE   | MAE    | MRE   | STD  |
|---------|------|-------|------|--------|--------|-------|------|
| BC      | 1.00 | 0.00  | 1.00 | 0.00   | 0.00   | 0.4%  | 0.00 |
| BrC     | 1.00 | 0.00  | 1.00 | 0.00   | 0.00   | 2.7%  | 0.00 |
| FNAI    | 1.02 | -0.02 | 0.99 | 0.03   | -0.01  | -1.0% | 0.03 |
| FNAS    | 1.03 | -0.03 | 1.00 | 0.01   | -0.02  | -6.0% | 0.01 |
| FAWC    | 0.99 | 0.00  | 1.00 | 0.02   | 0.00   | -0.2% | 0.02 |
| RH      | 0.94 | 0.04  | 0.97 | 0.03   | 0.00   | 0.3%  | 0.03 |
| CAI     | 1.00 | 0.00  | 1.00 | 0.00   | 0.00   | -1.1% | 0.00 |
| CNAI    | 0.95 | 0.01  | 0.99 | 0.02   | 0.00   | 8.2%  | 0.02 |
| CNAS    | 0.95 | 0.00  | 1.00 | 0.01   | -0.02  | -4.5% | 0.01 |
| CAWC    | 1.01 | 0.00  | 1.00 | 0.02   | 0.00   | 0.9%  | 0.02 |
| AOT     | 1.00 | 0.00  | 1.00 | 0.00   | 0.00   | 0.0%  | 0.00 |
| SSA     | 1.00 | 0.00  | 1.00 | 0.00   | 0.00   | 0.0%  | 0.00 |
| FM($n$) | 0.98 | 0.03  | 1.00 | 0.00   | 0.00   | 0.1%  | 0.00 |
| FM($k$) | 1.00 | 0.00  | 1.00 | 0.0003 | 0.0001 | 0.5%  | 0.00 |
| CM($n$) | 1.00 | 0.00  | 1.00 | 0.00   | 0.00   | 0.0%  | 0.00 |
| CM($k$) | 0.96 | 0.00  | 1.00 | 0.0000 | 0.0000 | 5.8%  | 0.00 |






**Table 3.** The statistics of aerosol parameters in Fig. 10: number of measurements (N), slope (A), intercept (B), correlation coefficient (R), root-mean-square error (RMSE), mean absolute error (MAE), standard error deviation (STD). GRASP approach (GA): Maxwell-Garnett (MG) mixing model, volume-weighted (VW) mixing model; standard (ST) GRASP/PARASOL retrievals without aerosol composition mixing model.

| | Banizoumbou AOT (675 nm) | | | Tamanrasset AOT (675 nm) | | | Mongu AOT (675 nm) | | |
|------|------|------|------|------|------|------|------|------|------|
| N | 78 | | | 76 | | | 118 | | |
| GA | MG | VW | ST | MG | VW | ST | MG | VW | ST |
| A | 0.75 | 0.96 | 0.68 | 0.68 | 0.88 | 0.49 | 0.90 | 0.96 | 0.96 |
| B | -0.02 | -0.05 | 0.08 | 0.03 | 0.06 | 0.13 | -0.01 | 0.00 | 0.00 |
| R | 0.97 | 0.96 | 0.91 | 0.89 | 0.88 | 0.51 | 0.96 | 0.95 | 0.94 |
| RMSE | 0.08 | 0.11 | 0.13 | 0.05 | 0.07 | 0.12 | 0.04 | 0.05 | 0.06 |
| MAE | -0.15 | -0.07 | -0.08 | -0.02 | 0.04 | 0.05 | -0.04 | -0.01 | -0.01 |
| STD | 0.13 | 0.11 | 0.19 | 0.07 | 0.07 | 0.14 | 0.05 | 0.05 | 0.06 |

| | Skukuza AOT (675 nm) | | | Solar village AOT (675 nm) | | | Agoufou AOT (675 nm) | | |
|------|------|------|------|------|------|------|------|------|------|
| N | 92 | | | 98 | | | 117 | | |
| GA | MG | VW | ST | MG | VW | ST | MG | VW | ST |
| A | 0.83 | 0.96 | 0.89 | 0.75 | 0.83 | 0.67 | 0.83 | 0.98 | 0.72 |
| B | -0.01 | 0.00 | 0.01 | 0.00 | 0.02 | 0.10 | 0.00 | 0.00 | 0.20 |
| R | 0.79 | 0.76 | 0.84 | 0.91 | 0.91 | 0.79 | 0.94 | 0.94 | 0.84 |
| RMSE | 0.05 | 0.06 | 0.04 | 0.09 | 0.10 | 0.13 | 0.14 | 0.16 | 0.21 |
| MAE | -0.03 | -0.01 | -0.01 | -0.11 | -0.06 | -0.05 | -0.10 | -0.01 | 0.04 |
| STD | 0.05 | 0.06 | 0.04 | 0.11 | 0.11 | 0.16 | 0.16 | 0.16 | 0.25 |

| | All sites AOT (675 nm) | | | All sites AE (870/440) | | | All sites SSA (675 nm) | | |
|------|------|------|------|------|------|------|------|------|------|
| N | 579 | | | 429 | | | 101 | | |
| GA | MG | VW | ST | MG | VW | ST | MG | VW | ST |
| A | 0.79 | 0.93 | 0.77 | 0.86 | 0.79 | 0.88 | 0.57 | 0.59 | 0.65 |
| B | 0.00 | 0.00 | 0.07 | 0.20 | 0.17 | 0.16 | 0.44 | 0.42 | 0.32 |
| R | 0.95 | 0.95 | 0.88 | 0.93 | 0.92 | 0.94 | 0.83 | 0.84 | 0.77 |
| RMSE | 0.09 | 0.11 | 0.15 | 0.24 | 0.24 | 0.24 | 0.02 | 0.02 | 0.03 |
| MAE | -0.07 | -0.02 | -0.01 | 0.08 | 0.00 | 0.06 | 0.04 | 0.04 | 0.00 |
| STD | 0.11 | 0.11 | 0.17 | 0.26 | 0.29 | 0.25 | 0.03 | 0.03 | 0.04 |



**Figures**

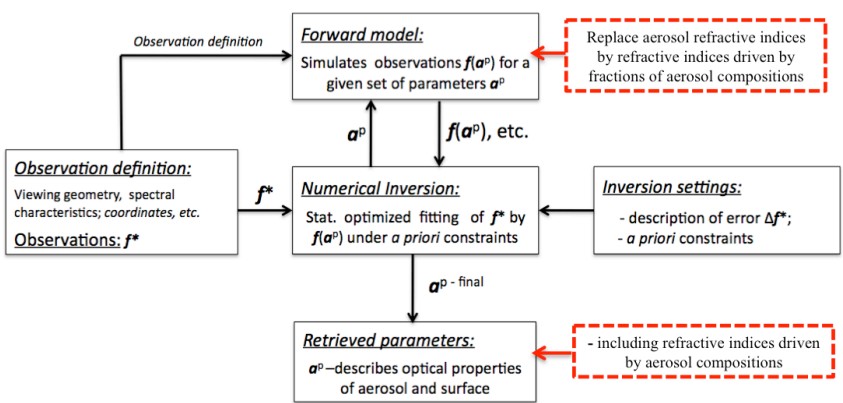

**Figure 1.** The general structure of GRASP algorithm with aerosol composition conversion
model, courtesy of (Dubovik et al., 2011). The red dashed frames represent modifications
for the composition inversion approach. $f^*$ represents vector of inverted measurements, $a^P$
represents vector of unknowns at the $p$-th iteration, $f(a^P)$ represents vector of measurement
fit at the $p$-th iteration.

**Figure 2.** Illustrates a general logistics of an effective refractive index calculation using a
conversion model that is based upon the Maxwell-Garnett effective medium approximation.



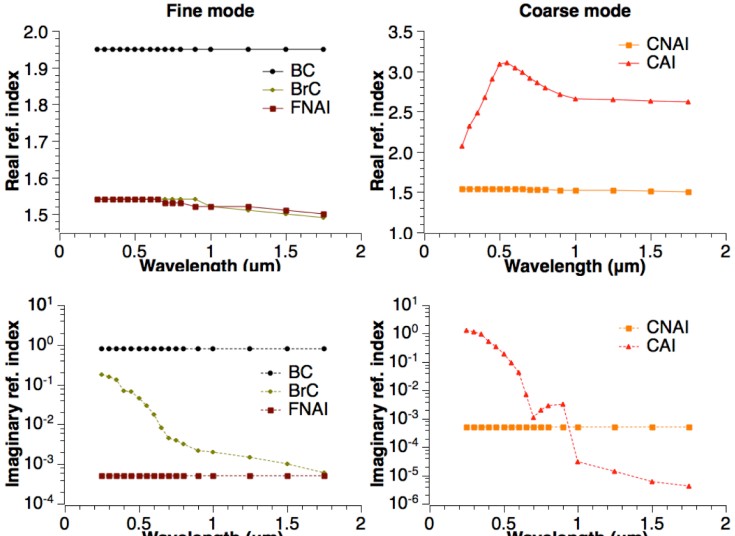

**Figure 3.** The refractive indices of assumed aerosol compositions embedded in the host of the
size-dependent Maxwell-Garnett conversion model. The parameters of BC refer to Bond and
Bergstrom (2006). The parameters of BrC refer to Sun et al. (2007) and Schuster et al. (2016).
The parameters of fine non-absorbing insoluble (FNAI) and coarse non-absorbing insoluble
(CNAI) refer to Ghosh (1999). FNAI represents dust and OC in fine mode particles, while
CNAI represents dust in coarse mode particles. The parameters of coarse absorbing insoluble
(CAI) refer to Longtin et al. (1988) representing hematite.

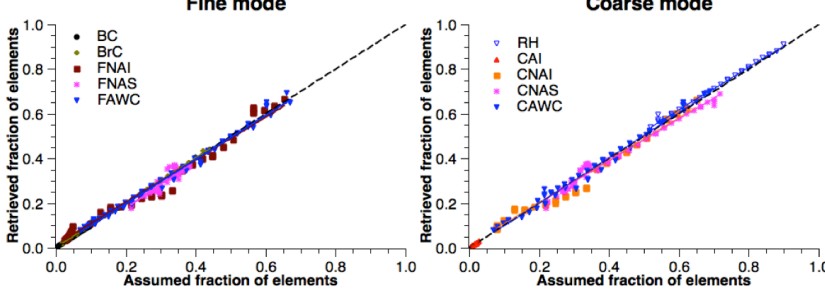

**Figure 4**. Assumed and retrieved fractions of aerosol composition species resulting from the
sensitivity tests of GRASP composition retrieval using Maxwell-Garnett mixing model.



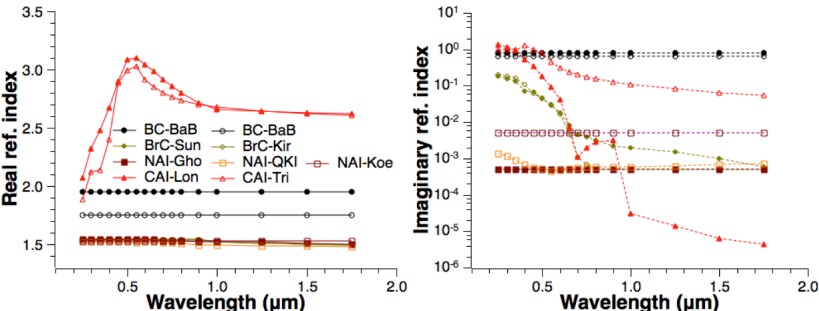

**Figure 5**. Complex refractive index of several aerosol species (BC, BrC, CAI, and NAI) in the host. The values with filled symbols are used in the presented size-dependent Maxwell-Garnett conversion model. The values with open symbols are used to replace the corresponding values to test the uncertainties in the aerosol composition retrievals. "BaB" for Bond and Bergstrom (2006); "Sun" for Sun et al. (2007); "Kir" for Kirchstetter et al. (2004); "Gho" for Ghosh (1999); "QKI" for dust composed of a mixture of quartz (Ghosh, 1999), kaolinite (Sokolik and Toon, 1999) and illite (Sokolik and Toon, 1999) with the proportions of 48%, 26%, and 26%, respectively (Journet et al., 2014); "Koe" for Koepke et al. (1997); "Lon" for Longtin et al. (1988); and "Tri" for Triaud (2005).

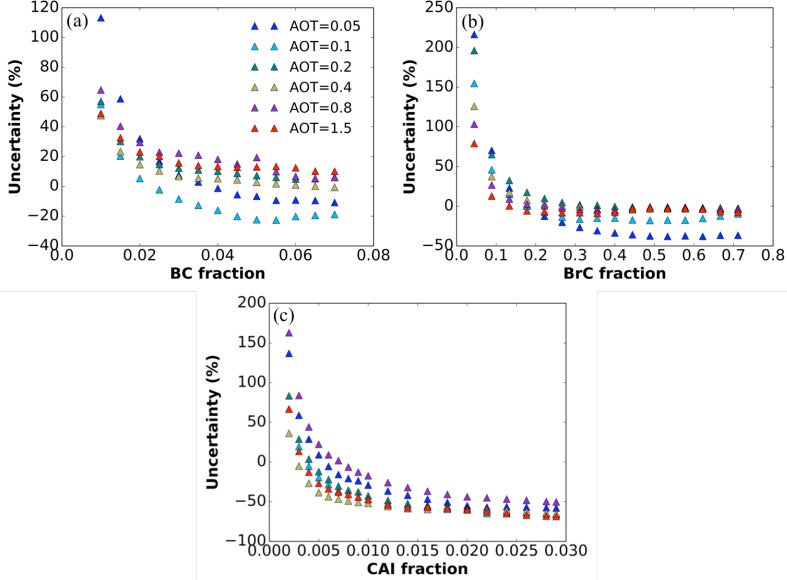

**Figure 6.** Uncertainty in absorbing species retrievals from POLDER/PARASOL attributed to the refractive index variability; uncertainties in (a) BC, (b) BrC, and (c) CAI fractions.



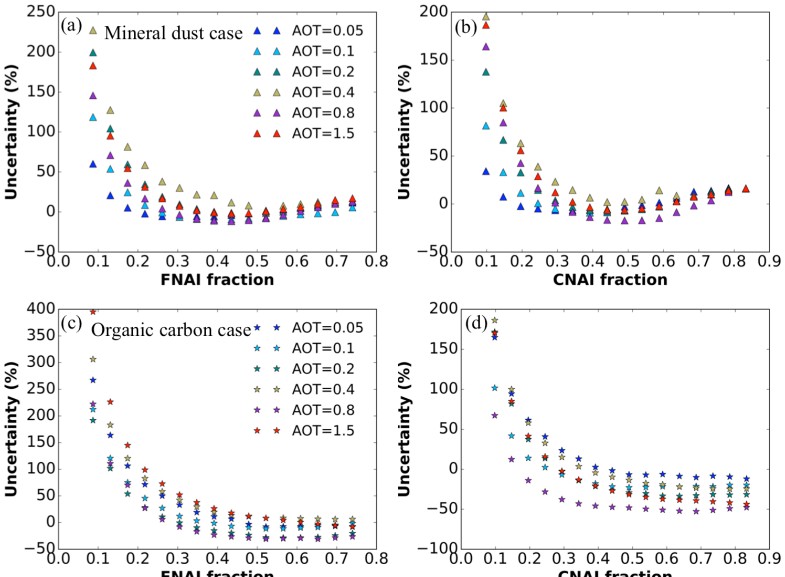

**Figure 7.** Uncertainty in Non-Absorbing Insoluble particles fraction in Fine (FNAI) and Coarse (CNAI) modes attributed to the refractive index variability: (a), (b) for the case of mineral dust and (c), (d) for organic carbon.

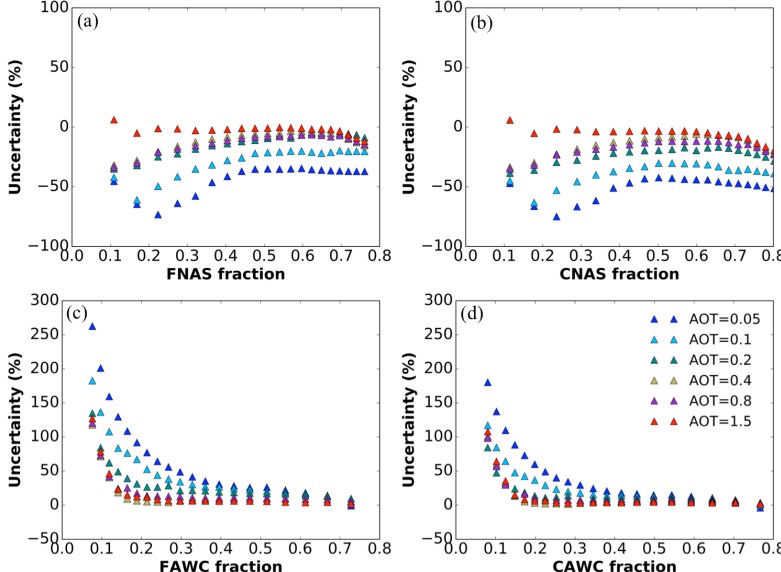

**Figure 8.** Uncertainty in Non-Absorbing Soluble particles and aerosol water content fraction in Fine (FNAS, FAWC) and Coarse (CNAS, CAWC) modes attributed to the refractive index and hygroscopic properties of ammonium nitrate and ammonium sulfate.

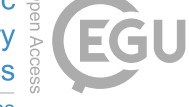


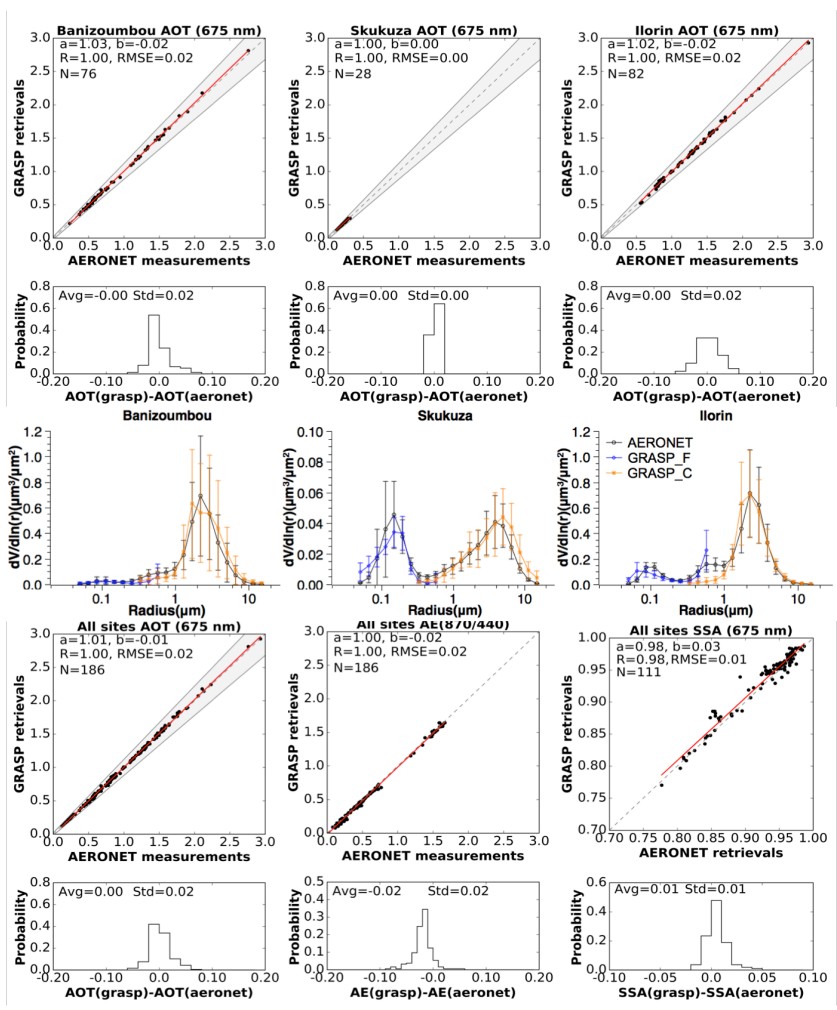


**Figure 9.** The inter-comparison of aerosol optical properties derived from Sun/sky photometer measurements using the GRASP/Composition approach with the corresponding values of the AERONET operational product. The data presented for the Banizoumbou site in April 2007 represent mineral dust aerosol, for the Skukuza site in September 2007 represent the biomass burning aerosol, and for the Ilorin site in January 2007 represent the mixture of dust and biomass burning.

1798
1799





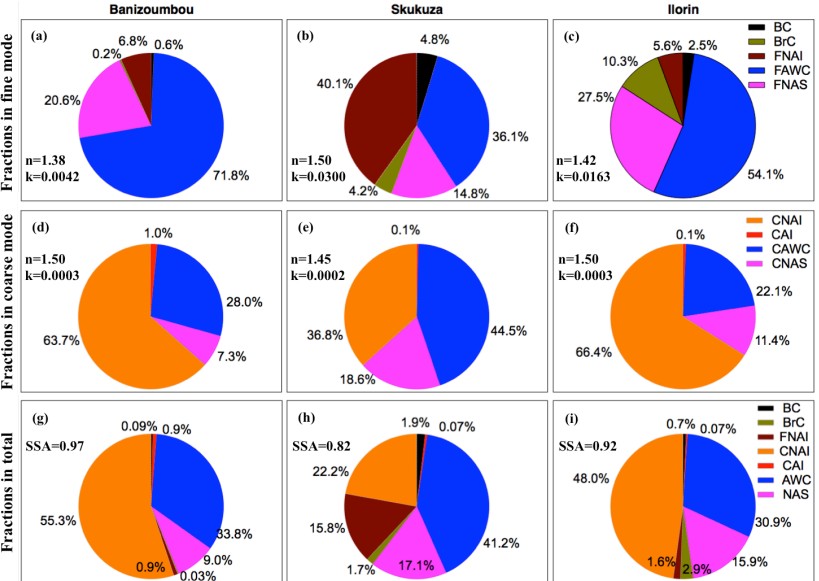

**Figure 10**. Examples of aerosol composition retrievals derived from AERONET Sun/sky photometer measurements using the GRASP/Composition approach. Panels: (a, d, g) the mineral dust case at the Banizoumbou site (April 8[th], 2007); (b, e, h) the biomass burning case at the Skukuza site (September 2[nd], 2007); and (c, f, i) the mixture of dust and biomass burning at the Ilorin site (January 25[th], 2007). In the panes are also indicated the values of complex refractive index (n, k) at 675 nm retrieved for the fine and coarse modes, and of SSA at 675 nm derived for ensemble of aerosol.





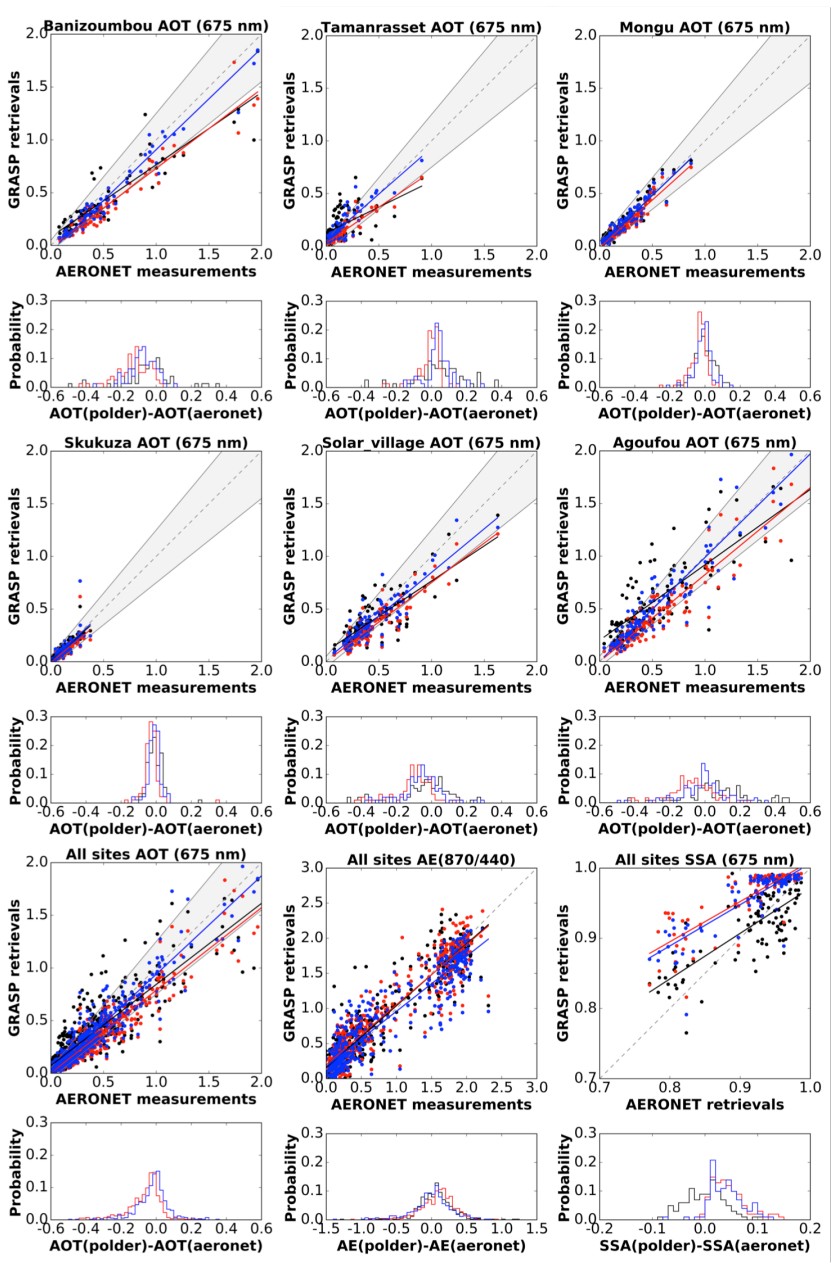

**Figure 11**. Inter-comparison of aerosol optical properties retrieved from POLDER/PARASOL and provided by AERONET operational product in six AERONET sites located in Africa and Middle East during the period 2006 to 2008. Red color represents the Maxwell-Garnett (MG) mixing model; blue color represents the volume-weighted (VW) mixing model; and black color represents the standard (ST) GRASP/PARASOL product that do not employ the aerosol composition retrievals.






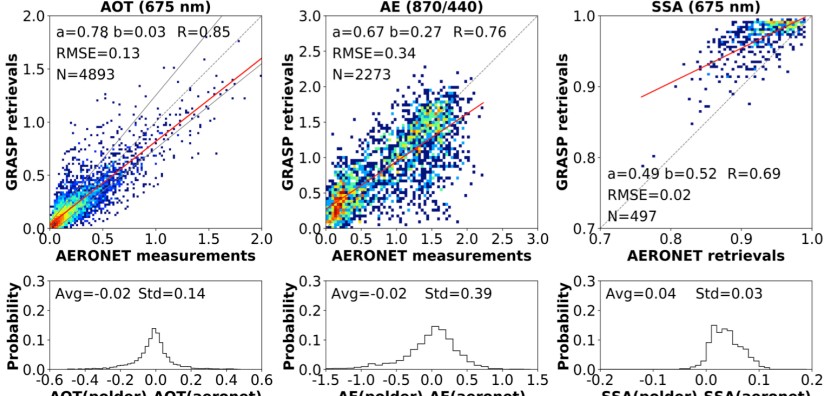

**Figure 12.** Inter-comparison of aerosol optical properties retrieved using the
POLDER/PARASOL composition (MG mixing model) approach and the corresponding
operational AERONET products from all globally available sites in 2008.



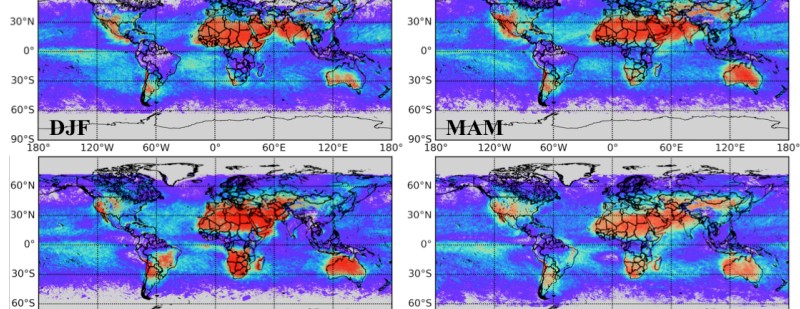

**Figure 13**. Seasonal variability of number of pixels in 0.1×0.1 degree resolution observed by
POLDER/PARASOL satellite over the globe in 2008.





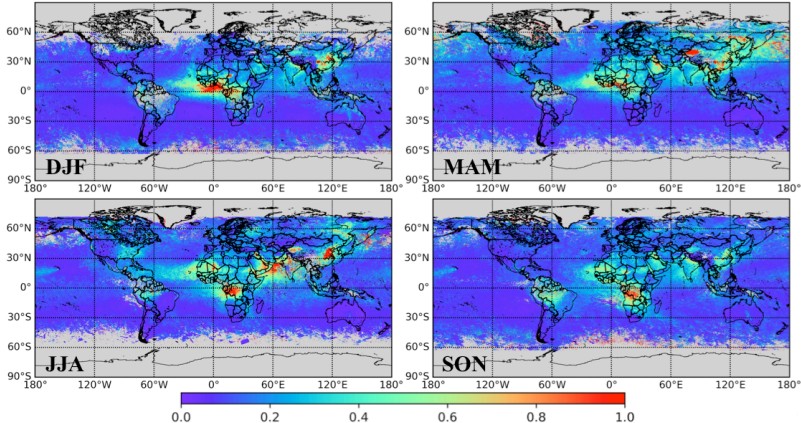

**Figure 14.** Seasonal variability of AOT at 565 nm in 2008 as retrieved by GRASP/Composition algorithm from POLDER/PARASOL satellite observations.

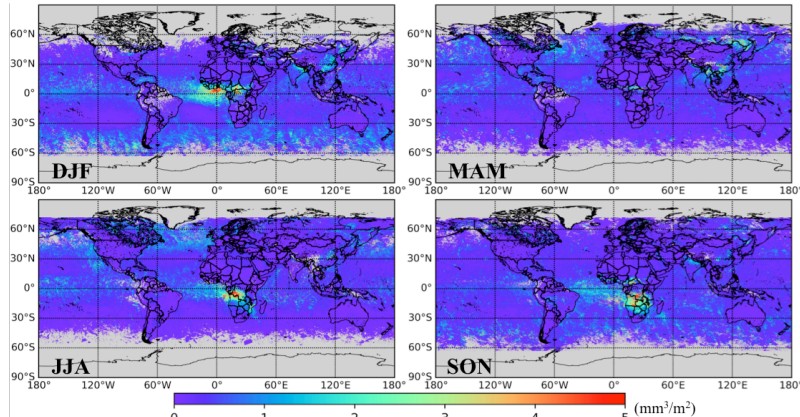

**Figure 15.** Seasonal variability of BC column volume concentration $(mm^3/m^2)$ over the globe in 2008 as retrieved by GRASP/Composition algorithm from POLDER/PARASOL satellite observations.





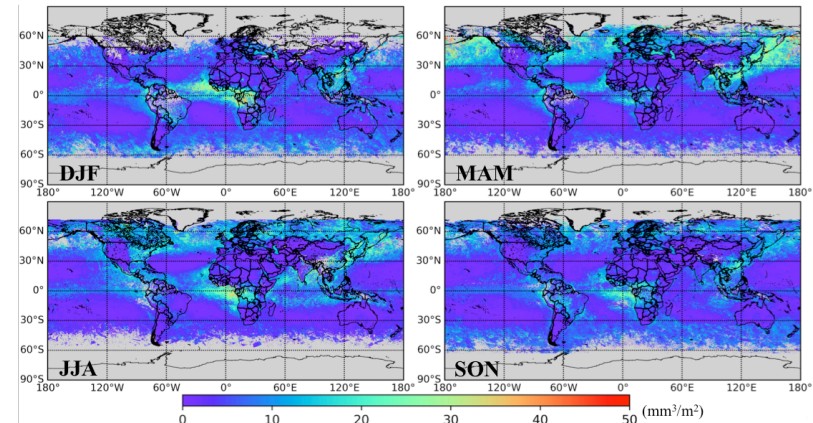

**Figure 16.** Same as Fig. 15, but for BrC

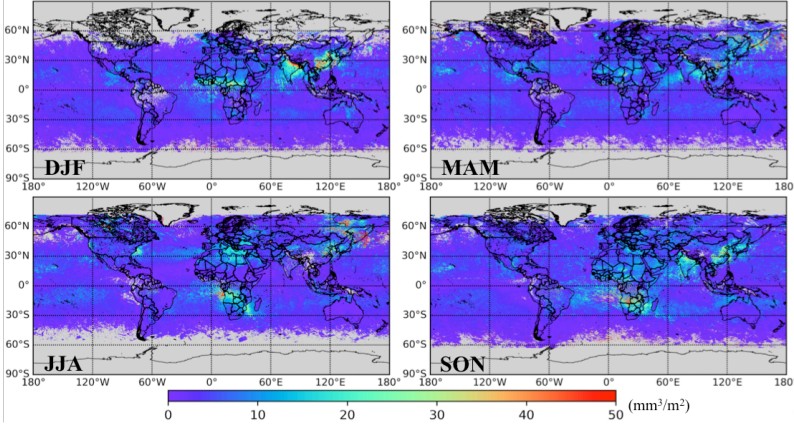

**Figure 17**. Same as Fig. 15, but for fine mode non-absorbing soluble (FNAS)





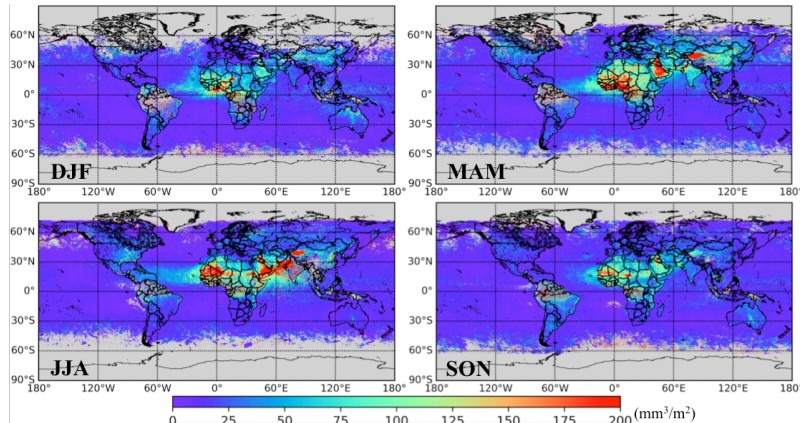

**Figure 18**. Same as Fig. 15, but for coarse mode non-absorbing insoluble (CNAI, dust)

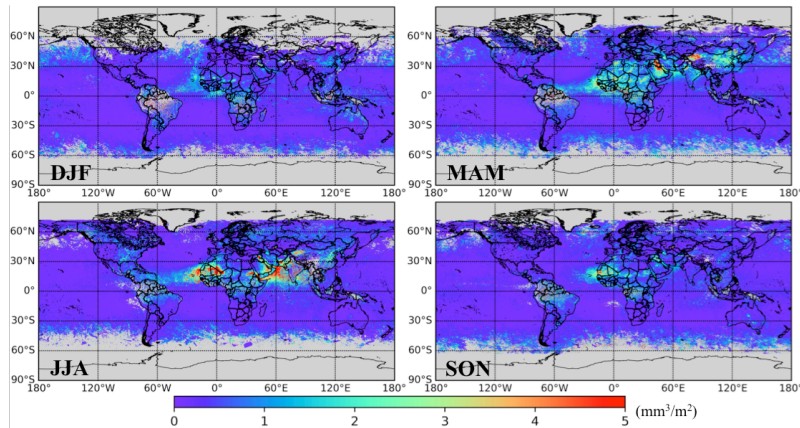

**Figure 19**. Same as Fig. 15, but for coarse mode absorbing insoluble (CAI, FeOx and Carbonaceous)



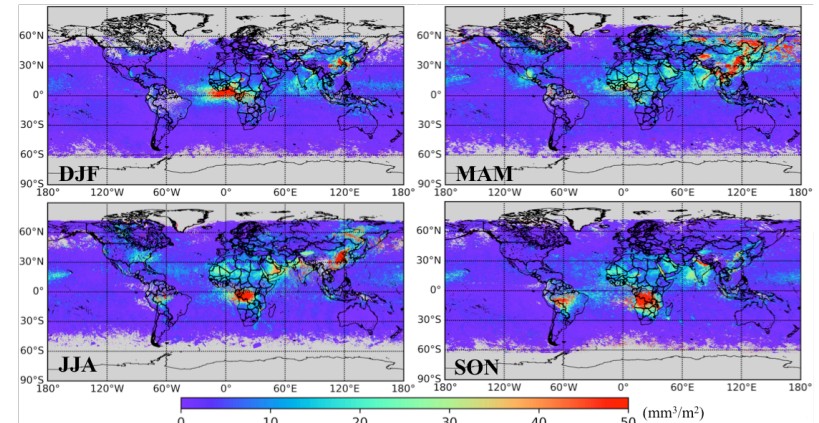

**Figure 20**. Same as Fig. 15, but for fine mode non-absorbing insoluble (FNAI, dust and OC)