# Peer review of "Retrieval of aerosol component directly from satellite and ground-based measurements"

_Atmospheric Chemistry and Physics, 2019_

## Referee Comment (RC1) · Anonymous Referee #1 · 9 May 2019

This paper presents an interesting strategy for using polarized/multiangular satellite measurements to infer aerosol composition. The method uses the Generalized Retrieval of Aerosol and Surface Properties (GRASP) along with assuming that aerosols are mixtures of non-soluable particles embedded within a soluable host. GRASP then derives size distribution, loading and light absorption characteristics based on determining the fractions of each aerosol type. The algorithm is applied first on synthetic data, then on ground-based AERONET, then on historic POLDER data.

Although quite long, I find this paper to be well-organized, well-written and worthy of publication. I really like the idea that the GRASP retrieval can combine specific aerosol types (size + shape + absorption) to retrieve the aggregate. This makes it possible to compare directly with models (that also assume these types). I think the authors should

highlight this even more than they have already. Finally, this is probably well beyond the scope of the paper, But I would like to see some sort of graphic (even for a single panels from Figs 15-19), to compare how your global maps (of species type) compare with other maps, such as from AeroCom models, and/or MISR size/shape/absorption climatology. Right now, they look reasonable, but I would be curious whether they might "change" our way of thinking about aerosol types distribution.

I have some small suggestions for improving the paper. They are as follows

1) Specifically, what are the aerosol type/model components? I feel as if a table could be used to describe each component, its size/shape and refractive index components.

2) Around line 290, the terms Fraci(i) and dV/dlnr appear without definition. I guess they are in the table 1, but since table 1 (in the PDF) wasn't near the text, one might want to define first time in the text.

3) The stars on equation (1) are confusing. Do the stars represent apriori or solutions?
4) In line 355, Csph appears (see comment #2).

5) How are the intrinsic aerosol parameters allowed to vary in time and space?

6) For Eq (7), there appear to be a lot of zeros in the matrix. We can assume there is no covariance? For example, I can't see avc (volume size distribution) and av (total volume) as being independent

7) Line 396-397: I guess I am curious, what do you mean by: "choice of mixing rule.. significantly affects the results". Can you show something about this?

8) It appears that Eq (11) and Eq (12) have the same RHS?

9) Line 448 looks like a formula (11 minus 12) not 11 and 12.

10) Lines 455-484: See my comment #1.

11) I have a few comments regarding figure 5 and paper text. Could Fig 5a be split into

two? There is a huge range of real refractive indices for CAI, but not for BrC and NAI. I cannot tell if the differences in assumptions for BrC and NAI around 1.5 are significant.

12) Line 532-540. These range of values could also be added to a table (e.g. #1)

13) Line 545-546: "Elevated" meaning larger loadings or higher altitudes?

14) Lines 547-550: Are the fractions of BC and CAI somehow constrained so they can't be "large"? I note that they never approach 0.5 and barely approach 0.1

15) For the plots of Figure 6, I am wondering what the "uncertainty" is. Should I read this as Uncertainty is fraction (%) of fraction? What if these were presented in same units as x-axis (fraction)? Of course estimates of tiny fractions should have large % uncertainties, (but then that also means that the estimates of the fraction of the other elements will have lower % uncertainties).

16) Line 581. I don't understand: "The non-absorbing insoluble can stand also for the insoluble organic carbon"

17) Some of the figures have panels with cut-off axes (e.g. Fig 3)

18) Are the units Fig 15-19 correct? (mm3/m2)?

---

## Referee Comment (RC2) · Anonymous Referee #2 · 2 Jun 2019

This paper describes a modification to the GRASP algorithm to generate retrievals of a predefined basis set of aerosol "species." Typically the GRASP and the Dubovik and King (hereafter DK) provide a retrieval of size and real/complex refractive index. This is then used by others to classify species. Here they tie the refractive indices retrieved by GRASP by a basis set aerosol species. Their primary point is that using a predefined basis set of species is more direct than inferring typing from the measured extinction and subsequently derived absorption angstrom exponent method (or in a few cases adding index of refraction) commonly used in the community. As I finished this review, I did do a quick comparison with reviewer 1. I would agree with their point that perhaps the most important aspect of this work is it provides something of a forward operator to perform more apples to apples comparisons between satellite and models to close

the radiance fields. I also agree that discussion of this point should be expanded. This discussion needs to cover these points.

I think this effort moves the field forward in that they change the basis set as to what the retrieval is producing (BC, BrC, soluble and insoluble). However, these parameters are by no means equivalent to "composition," which is their premise for the entire paper. "Soluble and insoluble fractions" are wholly ill-defined in the context of composition, as they are related really to hygroscopicity. This is also true in part in regard to BC and BrC as they were functionally optical parameters long before we knew much of their true chemical nature. Here they are an indicator of spectral absorption properties. There are a myriad of soluble species with different indices of refraction and hygroscopic properties and likewise spectral dependencies of absorption based on mixtures. Thus, the idea that what is being retrieved as independent information on composition is fundamentally not true. What is a step forward, is they demonstrated that using the GRASP algorithm, you can generate a retrieval of parameters other than the standard size, index of refraction etc. as a basis of some categorization of the optical environment. One could look at this as a complex transform, but really it is simply a way of having an a priori set of basis functions for different aerosol species. But this is sort of what most traditional aerosol retrievals do, provide a best fit on the developers notion of what the aerosol environment looks like. Coarse mode dust, fine mode pollution, absorbing components, etc. So when the authors say they are the first ones to extract composition directly from satellite, this is not true either, it is just the first time in the GRASP algorithm has taken this approach. Rather, they are taking some liberty with the language. My first major comment is I think the authors need to be very careful about their language here. They should be more up front as to what they are doing, or spend much more time explaining why they think there is something fundamentally different in their approach. My second point is that it is unclear as to what is really going on in the retrieval. If one considers the GRASP algorithm simply has a smoothness of fit contain, but otherwise can pick any refractive index and size it wants, why not just take the standard GRASP algorithm as it is, and after the fact

match the size and refractive index to any basis function they like? Or is that what they are already doing? (it is hard to tell in figure 1 and the associated discussion). Is the goodness of fit always the same from the standard GRASP algorithm, or does your predefinition of species leave a residual? If so, how big is that residual? They also list as an example the refractive index for ammonium nitrate, but what of the other species? We are referred to a GACP dataset, and Figure 3 has such dynamic range between species it is hard to tell even what these values are. I think an appendix needs to be generated that provides details on these key aspects of the retrieval. I think in order for them to prove validity of the algorithm, they should do the retrieval with their basis set, and then as a baseline compare to the standard grasp algorithm, and see to what extent the goodness of fit to the radiance fields changes.

My final major comment is that there is really very little verification work provide that shows that the results of the retrievals are fundamentally better than other categorization methods (that is getting back to the question in point 2 (if this method leaves a residual from the free running GRASP algorithm). Or can you baseline against a simple AE vs AAE plot for species? One could argue that real verification has always been an issue for retrievals. There are many studies that show that DK retrievals provide reasonable results. But, those studies and here pick sites that are generally single aerosol specie dominated (the once exception is Solar village). This is why at least a self-consistent baseline against the standard GRASP and DK retrievals is so important. The discussions in Section 4 related to Figure 14-20 global maps that provide some truthiness. But close examination (which required me zooming way into the plots) shows many logical inconsistencies, especially around coastlines, where the overall hydration of the particles leads to an increase in the "soluble fraction" Likewise there is a great deal of "insoluble" AOD in Brazil-even though we know smoke organic components do in fact have a hygroscopicity to them, even if it is low), as well as retrieval errors. Are they not really just applying a form Schuster's "water fraction" algorithm? This then closes the loop with comment 1: are they really doing a soluble and insoluble aerosol specie? In the end, I appreciate what the authors are trying to do, and can see

how this will benefit the community. But there are logic issues that need to be wrung out, and some form of baseline verification that shows this method is actually taking us in the right direction.

---

## Author Comment (AC1) · 19 Jul 2019

Response to comments of referee #1 on "Retrieval of aerosol composition directly from satellite and ground-based measurements" by Lei Li et al.

We appreciate the referee's thoughtful reading, valuable suggestions and time that we hope helped us to improve the manuscript. Our point-to-point replies are presented below.

Anonymous Referee #1 This paper presents an interesting strategy for using polarized/multiangular satellite measurements to infer aerosol composition. The method uses the Generalized Retrieval of Aerosol and Surface Properties (GRASP) along with assuming that aerosols are mixtures of non-soluble particles embedded within a sol-

[Figure]

uble host. GRASP then derives size distribution, loading and light absorption characteristics based on determining the fractions of each aerosol type. The algorithm is applied first on synthetic data, then on ground-based AERONET, then on historic POLDER data.

General comments: Although quite long, I find this paper to be well-organized, well-written and worthy of publication. I really like the idea that the GRASP retrieval can combine specific aerosol types (size + shape + absorption) to retrieve the aggregate. This makes it possible to compare directly with models (that also assume these types). I think the authors should highlight this even more than they have already. Finally, this is probably well beyond the scope of the paper, But I would like to see some sort of graphic (even for a single panels from Figs 15-19), to compare how your global maps (of species type) compare with other maps, such as from AeroCom models, and/or MISR size/shape/absorption climatology. Right now, they look reasonable, but I would be curious whether they might "change" our way of thinking about aerosol types distribution. Response: We appreciate the referee's suggestion. The direct bridging to the models is highlighted now in the abstract and the discussion is extended in the manuscript. Also, in fact, a comparison of our GRASP retrievals with GEOS-5/GOCART model estimations was already done. However, as the referee mentioned, this next step is beyond the scope of the current publication. Either this observation-based aerosol components maps will "change" or not our way of thinking about aerosol types distribution is highly interesting and desires a more focused effort. Nevertheless, to give a glance on how the retrievals agree with the models, a couple of additional figures are presented below. Figure R1 shows the correlations for monthly averaged dust mass concentrations over the globe for 2008. It shows, for example that the global average coarse dust mass concentration retrieved by GRASP (125.5 mg/m2) is lower than that simulated by GEOS-5/GOCART (174.7 mg/m2). Figure R2 shows the map of coarse mode dust distribution in April 2008. The retrievals tend to provide lower coarse dust concentration over the mineral dust region (such as in Northern Africa and Northern China), which is consistent with the previous studies showing that models

overestimate mineral dust in dust-dominated places (Jones and Christoper, 2007; Gi-noux et al., 2001, Chin et al., 2009). Also, the observations show quite a fine spatial distribution structure.

Figure R1. Correlation between the monthly averaged dust mass concentrations of GRASP retrievals and GEOS-5/GOCART model estimations over the globe in 2008.

Figure R2. Distribution of dust mass concentration estimated by GEOS-5/GOCART model and retrieved by GRASP/POLDER in April 2008.

Chin, M., Diehl, T., Dubovik, O., Eck, T. F., Holben, B. N., Sinyuk, A., and Streets, D. G.: Light absorption by pollution, dust, and biomass burning aerosols: a global model study and evaluation with AERONET measurements, Ann. Geophys., 27, 3439-3464, 2009. Ginoux, P. Chin, M., Tegen, I., Prospero, J. M., Holben, B., Dubovik, O., and Lin, S.-J.: Sources and distributions of dust aerosols simulated with the GOCART model, 106(D17), 20255-20273, 2001. Jones, T. A. and Christopher, S. A: MODIS derived fine mode fraction characteristics of marine, dust, and anthropogenic aerosols over the ocean, constrained by GOCART, MOPITT, and TOMS, J. Geophys. Res., 112(D22), 2007.

Specific comments: some small suggestions for improving the paper. 1) Specifically, what are the aerosol type/model components? I feel as if a table could be used to describe each component, its size/shape and refractive index components. Response: This useful suggestion is addressed by presenting a table summarizing the aerosol components description. The complex refractive indices of each component are presented at 0.440 ðÌIJĞðÍŚŽ and 0.865 ðÌIJĞðÍŚŽ.

2) Around line 290, the terms Frac(i) and dV/dlnr appear without definition. I guess they are in the table 1, but since table 1 (in the PDF) wasn't near the text, one might want to define first time in the text. Response: Thank you for noticing this. We added the definitions of "Frac(i) and $(dV(r\_j))/dlnr$ (i=1,...,N\_r)" in the text.

3) The stars on equation (1) are confusing. Do the stars represent a priori or solutions? Response: The term with a star represents satellite measurements. For example, $f\_i\hat{}\,^*$ denotes a vector of the measurements and $f\_i$ denotes a vector of the estimations. It is now clarified in the text.

4) In line 355, Csph appears (see comment #2). Response: The $C_{(sph)}$ definition was added in the text.

5) How are the intrinsic aerosol parameters allowed to vary in time and space? Response: Besides the aerosol component retrieval module, the presented algorithmic developments essentially rely on the available heritage of the previous retrieval developments for POLDER space instrument. Thus, as in the standard GRASP algorithm (Dubovik et al., 2011), the satellite retrieval is designed as a statistically optimized simultaneous fitting of the observations over a group of pixels implemented under additional inter-pixel constraints. Specifically, the variations of the retrieved parameters horizontally from pixel-to-pixel or temporary from day-to-day over the same pixel are limited by the additional a priori constraints, in a similar manner to how it is applied in inverse modeling by Dubovik et al. (2008). The inclusion of these additional constraints is expected to provide retrieval of higher consistency for aerosol retrievals from satellites, because the retrieval over each single pixel will be benefiting from coincident aerosol information from neighboring pixels, in addition to the information about surface reflectance (over land) obtained in preceding and consequent observations over the same pixel.

Dubovik, O., Herman, M., Holdak, A., Lapyonok, T., Tanré, D., Deuzé, J.L., Ducos, F., Sinyuk, A., Lopatin, A.: Statistically optimized inversion algorithm for enhanced retrieval of aerosol properties from spectral multi-angle polarimetric satellite observations, Atmos. Meas. Tech., 4, 975–1018, https://doi.org/10.5194/amt-4-975-2011, 2011. Dubovik, O., Lapyonok, T., Kaufman, Y.J., Chin, M., Ginoux, P., Kahn, R.A., Sinyuk, A.: Retrieving global aerosol sources from satellites using inverse modeling, Atmos. Chem. Phys., 8, 209–250, https://doi.org/10.5194/acp-8-209-2008, 2008.

6) For Eq (7), there appear to be a lot of zeros in the matrix. We can assume there is no covariance? For example, I can't see avc (volume size distribution) and av (total volume) as being independent Response: Zeros in the matrix denote that there are no smoothness constraints for these parameters, such as the spectral smoothness constraint. A clarification sentence is added now in the text.

7) Line 396-397: I guess I am curious, what do you mean by: "choice of mixing rule…significantly affects the results". Can you show something about this? Response: This statement should be indeed clarified. The meaning is that the different assumptions on mixing rules, which are often used for estimation of aerosol composition, were reported by Xie et al. (2014) as affecting the aerosol composition retrieved by an independent algorithm that uses ground-based AERONET measurements. For instance, the Bruggeman approximation was found as more suitable for the dust case, the Maxwell-Garnett for the haze case, and volume average for the clean case. In our study, the Maxwell-Garnett and volume average mixing rules were employed independently of the aerosol type and the retrieval results are inter-compared (Figure 11). In our approach we have not identified a significant influence of the mixing rule choice on the quality of the retrievals. Moreover, the aerosol optical properties were rather well comparable in both cases. The fractions of the elements present some differences due to the differences in the formulation, but are still in a reasonable agreement. The text is modified accordingly.

Xie Y., Li Z., Li L. et al.: Study on influence of different mixing rules on the aerosol components retrieval from ground-based remote sensing measurements. Atmospheric Research, 145: 267-278, 2014.

8) It appears that Eq (11) and Eq (12) have the same RHS? Response: No, please note that there is a difference in the sign before $\varepsilon\_r$.

9) Line 448 looks like a formula (11 minus 12) not 11 and 12. Response: It is corrected.

10) Lines 455-484: See my comment #1. Response: A table summarizing description

of the aerosol components is added (see Table 2 in the revised manuscript and the response to comment #1).

11) I have a few comments regarding figure 5 and paper text. Could Fig 5a be split into two? There is a huge range of real refractive indices for CAI, but not for BrC and NAI. I cannot tell if the differences in assumptions for BrC and NAI around 1.5 are significant. Response: An axis break is added for the ordinate in Figure 5a to better show the different range of real refractive indices.

12) Line 532-540. These range of values could also be added to a table (e.g. #1) Response: The values of complex refractive indices at 0.440 ðĺIJĞð匎 and 0.865 ðĺIJĞð匎, used in the uncertainty test, are added to Table 2 (Please see the response to comment #1).

13) Line 545-546: "Elevated" meaning larger loadings or higher altitudes? Response: It is reworded to "Therefore, the estimates should be quite reasonable in the cases of large pollution loading."

14) Lines 547-550: Are the fractions of BC and CAI somehow constrained so they can't be "large"? I note that they never approach 0.5 and barely approach 0.1 Response: It is important to say that the maximal retrieved fractions for BC and CAI (mainly representing iron oxides) do not result from a constrain, despite have a limit due to the range of possible complex refractive indices in the pre-computed kernels of aerosol optical characteristics. That is, the volume fractions of these two highly absorbing species are indeed limited in the algorithm to 10% for BC and 3% for CAI, based on the below listed reasons. However, our analysis showed that these maximal values were never reached in the inversion procedure. Previous in-situ studies (Ganor and Foner, 1996; Guieu et al., 2002; Lafon et al., 2004, 2006, Alfaro et al., 2004; Wagner et al., 2012; Formenti et al., 2014) showed that iron oxides account for 2.8—6.5% of mineral dust by mass. It means approximately 1.4—3.25% by volume, since the density of free iron is much higher than other common minerals (4.28 g cm-3 for goethite and 5.25 g

cm-3 for hematite, as opposed to 2.65 g cm-3 for illite, kaolinite, quartz, and calcite; Formenti et al., 2014 The fraction of BC in atmospheric aerosol was generally reported not exceeding 10% (Bond et al., 2013). It is also to note that the retrievals of aerosol composition derived from AERONET measurements by Schuster et al. (2016) demonstrated that the volume fraction of free iron remains relatively constant in West Africa throughout the year (1.4—1.7%) and the volume fraction of black carbon reaches a peak of 1.0% for the fine mode during West African biomass burning season and a peak of 3.0% for the fine mode in southern Africa biomass burning. The corresponding precision is added to the manuscript (lines 590 to 609).

Alfaro, S., Lafon, S., Rajot, J., Formenti, P., Gaudichet, A., and Maille, M.: Iron oxides and light absorption by pure desert dust: an experimental study, J. Geophys. Res., 109, D08208, doi:10.1029/2003JD004374, 2004. Bond, T.C., Doherty, S.J., Fahey, D.W., Forster, P.M., Berntsen, T., Deangelo, B.J., Flanner, M.G., Ghan, S., Kärcher, B., Koch, D., Kinne, S., Kondo, Y., Quinn, P.K., Sarofim, M.C., Schultz, M.G., Schulz, M., Venkataraman, C., Zhang, H., Zhang, S., Bellouin, N., Guttikunda, S.K., Hopke, P.K., Jacobson, M.Z., Kaiser, J.W., Klimont, Z., Lohmann, U., Schwarz, J.P., Shindell, D., Storelvmo, T., Warren, S.G., Zender, C.S.: Bounding the role of black carbon in the climate system: A scientific assessment, J. Geophys. Res. Atmos., 118, 5380–5552, https://doi.org/10.1002/jgrd.50171, 2013. Formenti, P., Caquineau, S., Chevaillier, S., Klaver, A., Desboeufs, K., Rajot, J.L., Belin, S., Briois, V.: Dominance of goethite over hematite in iron oxides of mineral dust from Western Africa: Quantitative partitioning by X-ray absorption spectroscopy, J. Geophys. Res. Atmos., 119, 12740-12754, https://doi.org/10.1002/2014JD021668, 2014. Ganor, E. and Foner, H. A.: The mineralogical and chemical properties and the behavior of aeolian Saharan dust over Israel, in: The Impact of Desert Dust Across the Mediterranean, edited by: Guerzoni, S., and Chester, R., Kluwer Academic Publishers, Printed in the Netherlands, 163-172, 1996. Guieu, C., Loye-Pilot, M. D., Ridame, C., Thomas, C.: Chemical characterization of the Saharan dust end-member: Some biogeochemical implications for the western Mediterranean Sea, J. Geophys. Res., 107 (D15), 4258,

https://doi.org/4210.1029/2001JD000582, 2002. Lafon, S., Rajot, J. L., Alfaro, S. C., Gaudichet, A.: Quantification of iron oxides in desert aerosol, Atmos. Environ., 38, 1211-1218, 2004. Lafon, S., Sokolik, I.N., Rajot, J.L., Caquincau, S., Gaudichet, A.: Characterization of iron oxides in mineral dust aerosols: Implications for light absorption, J. Geophys. Res. Atmos., 111, 1–19, https://doi.org/10.1029/2005JD007016, 2006. Schuster, G.L., Dubovik, O., Arola, A.: Remote sensing of soot carbon - Part 1: Distinguishing different absorbing aerosol species, Atmos. Chem. Phys., 16, 1565–1585, https://doi.org/10.5194/acp-16-1565-2016, 2016. Wagner, R., Ajtai, T., Kandler, K., Lieke, K., Linke, C., Müller, T., Schnaiter, M., and Vragel, M.: Complex refractive indices of Saharan dust samples at visible and near UV wavelengths: a laboratory study, Atmos. Chem. Phys., 12, 2491–2512, doi:10.5194/acp-12-2491-2012, 2012.

15) For the plots of Figure 6, I am wondering what the "uncertainty" is. Should I read this as uncertainty is fraction (%) of fraction? What if these were presented in same units as x-axis (fraction)? Of course estimates of tiny fractions should have large % uncertainties, (but then that also means that the estimates of the fraction of the other elements will have lower % uncertainties). Response: Thank you for this comment, it is indeed not sufficiently explained in the text. The uncertainty is defined in percentage as (retrieved_fraction-assumed_fraction)/assumed_fraction. We would prefer to leave the notation in Figure 6 as it is, however, an explanation is added in the paragraph on uncertainty calculation (lines 646 to 647).

16) Line 581. I don't understand: "The non-absorbing insoluble can stand also for the insoluble organic carbon" Response: The sentence is reworded. The intention was to be mentioned in the text problematic that the non-absorbing dust and non-absorbing organic carbon have similarity in the complex refractive index and is not distinguishable for the employed type of observations. It is corrected to "The non-absorbing insoluble component can represent not only non-absorbing dust, but also non-absorbing organic carbon, as was mentioned above."

17) Some of the figures have panels with cut-off axes (e.g. Fig 3) Response: Corrected.

18) Are the units Fig 15-19 correct? (mm3/m2)? Response: It can be confusing, but the units are correct. The units denote the volume concentration in total atmospheric column with unit surface area. We clarified it in the text.

Please also note the supplement to this comment:
https://www.atmos-chem-phys-discuss.net/acp-2019-208/acp-2019-208-AC1-supplement.pdf
* * *
[Figure]

[Figure]

**Fig. 1.** Figure R1. Correlation between the monthly averaged dust mass concentrations of GRASP retrievals and GEOS-5/GOCART model estimations over the globe in 2008.

[Figure]

[Figure]

[Figure]

**Fig. 2.** Figure R2. Distribution of dust mass concentration estimated by GEOS-5/GOCART model and retrieved by GRASP/POLDER in April 2008.

**Table 2.** Description of aerosol components and complex refractive indices at 0.440 $\mu m$ and 0.865 $\mu m$ employed in the GRASP components retrieval approach, as well as those used in the uncertainty tests.

| Abb. | Component | Complex refractive index | | Reference |
|---|---|---|---|---|
| | | 0.440 $\mu m$ | 0.865 $\mu m$ | |
| BC | Black carbon representing wavelength-independent strong absorption | 1.95+0.79i | 1.95+0.79i | Bond & Bergstrom (2006) |
| | | 1.75+0.63i | 1.75+0.63i | Bond & Bergstrom (2006) |
| BrC | Brown carbon representing wavelength-dependent absorption | 1.54+0.07i | 1.54+0.003i | Sun et al. (2007) |
| | | 1.54+0.06i | 1.54+0.0005i | Kirchstetter et al. (2004) |
| FNAI | Fine mode non-absorbing insoluble representing fine non-absorbing dust and organic carbon | 1.54+0.0005i | 1.52+0.0005i | Ghosh (1999) |
| | | 1.53+0.005i | 1.53+0.005i | "GKI"[1] |
| | | 1.52+0.0006i | 1.50+0.0006i | Koepke et al. (1997) |
| FNAS | Fine mode non-absorbing soluble representing inorganic salts | $1.337+10^{-9}$i | $1.339+10^{-8}$i | Tang et al. (1981); Gosse et al. (1997) for "AN"[2] |
| | | $1.537+10^{-7}$i | $1.517+10^{-7}$i | Toon et al. (1976) for "AS" [3] |
| FAWC | Fine mode aerosol water content | $1.337+10^{-9}$i | $1.329+10^{-6.5}$i | Hale & Querry (1973) |
| CAI | Coarse mode absorbing insoluble representing iron oxides | 2.90+0.345i | 2.75+0.003i | Longtin et al. (1988) |
| | | 2.88+0.987i | 2.72+0.140i | Triaud (2005) |
| CNAI | Coarse mode non-absorbing insoluble represented by non-absorbing dust | 1.54+0.0005i | 1.52+0.0005i | Ghosh (1999) |
| | | 1.53+0.005i | 1.53+0.005i | "GKI"[1] |
| CNAI | by Organic Carbon | 1.52+0.0006i | 1.50+0.0006i | Koepke et al. (1997) |
| CNAS | Coarse mode non-absorbing soluble represented by an inorganic salt - AN[2] | $1.337+10^{-9}$i | $1.339+10^{-8}$i | Tang et al. (1981); Gosse et al. (1997) |
| CNAS | by AS[3] | $1.537+10^{-7}$i | $1.517+10^{-7}$i | Toon et al. (1976) |
| CAWC | Coarse mode aerosol water content | $1.337+10^{-9}$i | $1.329+10^{-6.5}$i | Hale & Querry (1973) |

"GKI"[1] denotes dust composed of a mixture of quartz (Ghosh, 1999), kaolinite (Sokolik and Toon, 1999) and illite (Sokolik and Toon, 1999) with the proportions recalculated from Journet et al. (2014).

"AN"[2] denotes ammonium nitrate, which can be used to create a host in aerosols.

"AS"[3] denotes ammonium sulfate, which is an alternative species for the host estimation in aerosols.

**Fig. 3.** Table 2. Description of aerosol components and complex refractive indices at 0.440 ðÍIJĞðÍŚŽ and 0.865 ðÍIJĞðÍŚŽ employed in the GRASP components retrieval approach, as well as those used in the uncertainty tests.

---

## Author Comment (AC2) · 19 Jul 2019

We appreciate the referee's time, constrictive and valuable comments that helped us to better consolidate and focus the manuscript. Our responses are presented below.

Anonymous Referee #2

General comments: This paper describes a modification to the GRASP algorithm to generate retrievals of a predefined basis set of aerosol "species." Typically the GRASP and the Dubovik and King (hereafter DK) provide a retrieval of size and real/complex refractive index. This is then used by others to classify species. Here they tie the refractive indices retrieved by GRASP by a basis set aerosol species. Their primary point is that using a predefined basis set of species is more direct than inferring typing

from the measured extinction and subsequently derived absorption angstrom exponent method (or in a few cases adding index of refraction) commonly used in the community. As I finished this review, I did do a quick comparison with reviewer 1. I would agree with their point that perhaps the most important aspect of this work is it provides something of a forward operator to perform more apples to apples comparisons between satellite and models to close the radiance fields. I also agree that discussion of this point should be expanded. This discussion needs to cover these points. I think this effort moves the field forward in that they change the basis set as to what the retrieval is producing (BC, BrC, soluble and insoluble). However, these parameters are by no means equivalent to "composition," which is their premise for the entire paper. "Soluble and insoluble fractions" are wholly ill-defined in the context of composition, as they are related really to hygroscopicity. This is also true in part in regard to BC and BrC as they were functionally optical parameters long before we knew much of their true chemical nature. Here they are an indicator of spectral absorption properties. There are a myriad of soluble species with different indices of refraction and hygroscopic properties and likewise spectral dependencies of absorption based on mixtures. Thus, the idea that what is being retrieved as independent information on composition is fundamentally not true. What is a step forward, is they demonstrated that using the GRASP algorithm, you can generate a retrieval of parameters other than the standard size, index of refraction etc. as a basis of some categorization of the optical environment. One could look at this as a complex transform, but really it is simply a way of having an a priori set of basis functions for different aerosol species. But this is sort of what most traditional aerosol retrievals do, provide a best fit on the developers notion of what the aerosol environment looks like. Coarse mode dust, fine mode pollution, absorbing components, etc. So when the authors say they are the first ones to extract composition directly from satellite, this is not true either, it is just the first time in the GRASP algorithm has taken this approach. Rather, they are taking some liberty with the language. My first major comment is I think the authors need to be very careful about their language here. They should be more up front as to what they are doing, or spend much more time explaining

why they think there is something fundamentally different in their approach. Response: We appreciate this very interesting review and valuable thoughts that pushed us to better communicate the novelty and originality of the proposed approach. First of all, we understand the importance and sensitivity to the semantic and agree that the word "composition" can be more associated with aerosol chemical composition, which does not properly reflect what is retrieved. However, to define as retrieval of aerosol type or classification will be wrong as well, as explained below. We therefore converged to aerosol "component" that will be used hereafter. We argue here and provide a discussion in the manuscript on why the suggested approach should be distinguished from the traditional aerosol retrievals (LUT approaches) and is fundamentally different to previous approaches. It should be noted that the retrieval of aerosol type has been clearly recognized as an important task by the scientific community and has been addressed in several studies. For example, there are a number of approaches that attempt to identify the type of aerosol through analysis of optical parameters such as single scattering albedo (SSA), Ångström Exponent (AE), AAE (absorption AE), refractive index, etc. Specifically, Russell et al. (2014) relate AERONET- and POLDER-derived optical properties to different aerosol types: urban, dust, marine, biomass burning, etc. Studies by Chung et al. (2010) and Bahadur et al. (2012) use AERONET optical properties like AE and AAE to separate BC, BrC, and dust into species-specific AAOT (absorption AOT). Schuster et al. (2005, 2009, 2016a) and Li et al. (2015) quantify the relative volume fractions of one or more aerosol species (e.g. BC, BrC, iron oxide, water) by adjusting the mixture of several components in an aerosol model to fit AERONET-retrieved refractive indices. However, our new approach differs substantially from all of these methods because it does not use a retrieval of optical parameters as an intermediate step. Thus, we expect the GRASP/Component approach to provide a stronger link to the radiation field than the previous approaches, as well as fundamentally higher retrieval accuracy. Moreover, some of above methods have additional differences and limitations compared to our proposed approach. For example, the Russell et al. (2014) approach is rather qualitative and does not attempt to quantify the relative volume or

mass fractions of different species in an aerosol mixture. Chung et al. (2012) and Bahadur et al. (2012) seem to use a technique for separating carbonaceous aerosols from dust that is not fully consistent with the AERONET retrieval assumptions, as discussed by Schuster et al. (2016b). Also, the Look-Up Table (LUT) approaches employed in most satellite retrievals (Martonchik et al., 1998; Remer et al., 2005; Kahn and Gaitley, 2015; Popp et al., 2016; Hammer et al., 2018; etc.) are designed to search amongst a preselected set of aerosol models (or their mixtures) for a model that provides the best fit to the observations. Since the models in a LUT are usually associated with a number of aerosol types (e.g. desert dust, smoke, urban aerosol etc.), the identification of the model that provides the best fit is often considered as a retrieval of aerosol type/composition. For observations with enhanced sensitivity, such as the Multi-angle Imaging SpectroRadiometer (MISR), a large number of models can be justified in the LUT and the differentiation of the models described by the ensembles of parameters can indeed be rather robust. However, LUT approaches are fundamentally limited to a discrete set of possible solutions, whereas the GRASP/Component approach searches through a continuous space of solutions; thus, the identification of aerosol components with our new methodology is significantly more detailed and elaborate. The proposed approach also bridges directly to the quantities of aerosol compositions used in the global chemical transport models. Specifically, our aerosol component retrievals can satisfy the requirements of chemical transport models to constrain their aerosol estimations on a large or global scale. However, we note that the GRASP/Component approach is only possible if 1) there is significant instrument sensitivity to the parameters that are related aerosol component (i.e. complex refractive index), and 2) this sensitivity is maintained while other parameters like the size distribution are adjusted. The retrieval provided in this article is unique because the aerosol component fractions are iterated in the original algorithm until a "best fit" is achieved with the measured radiance field, making it unnecessary to use retrieved optical properties as a constraint. Thus, the aerosol volume fractions determined with this new procedure have a direct link to the measured radiance field, whereas the procedures outlined above have a

direct link to the retrieved aerosol optical properties provided by AERONET or satellite products. In addition, the sensitivity of aerosol water fraction to the real refractive index is shown in Figure R1 (Schuster et al., 2009). Schuster et al. (2009) illustrated that the soluble aerosol components (sea salt, ammonium sulfate, ammonium nitrate, etc.) indicate similar refractive indices for similar mixing ratios, even though the dry refractive indices can be quite different. Hence, the aerosol water fraction can be derived from the mixture real refractive index if the aerosols are known to be one of the common soluble aerosols (in this study it is assumed as ammonium nitrate). Hygroscopicity, as an indicator of how aerosol growth in size or scattering, is related to relative humidity. However, in this study we are more concerned about how aerosol water content is related to real refractive index, which is much simpler (no hysteresis, no exponential growth, etc.) according to the study of Schuster et al. (2009) shown as Figure R1. These discussions are added to the manuscript.

Figure R1. Water fractions for two-component aerosol mixtures as a function of the real refractive index. (Schuster et al., 2009)

[revised manuscript text omitted]

My second point is that it is unclear as to what is really going on in the retrieval. If one considers the GRASP algorithm simply has a smoothness of fit contain, but otherwise can pick any refractive index and size it wants, why not just take the standard GRASP algorithm as it is, and after the fact match the size and refractive index to any basis function they like? Or is that what they are already doing? (it is hard to tell in figure 1 and the associated discussion). Is the goodness of fit always the same from the standard GRASP algorithm, or does your predefinition of species leave a residual? If so, how big is that residual? They also list as an example the refractive index for ammonium nitrate, but what of the other species? We are referred to a GACP dataset, and Figure 3 has such dynamic range between species it is hard to tell even what these values are. I think an appendix needs to be generated that provides details on these key

aspects of the retrieval. I think in order for them to prove validity of the algorithm, they should do the retrieval with their basis set, and then as a baseline compare to the standard grasp algorithm, and see to what extent the goodness of fit to the radiance fields changes. Response: Regarding the goodness of fit and validity of the new algorithm as compared to the standard GRASP algorithm: we thank for bringing up this point because it is indeed an essential step that was done in early stages of the new algorithm developments, but is not explicitly mentioned in the original manuscript. Figure 11 of the original version, however, already showed an inter-comparison of correlations between optical characteristics derived by the new approach, the standard GRASP and the operational AERONET product, which addresses in a way the raised question. Here, in Figure R2, we present the residuals from different GRASP approaches. The data subset used is the same as for the inter-comparison with the operational AERONET product (Figure 11 and Table 4). Figure R2 shows that the residuals of the GRASP/Component approaches (Maxwell-Garnett (MG) and volume weighted (VW) mixing rules) are almost the same as those of standard GRASP. Although the average residual of aerosol component approach (2.4±0.9% for MG and 2.4±1.0% for VW) is slightly higher than of standard GRASP (2.3±0.9%), the maximum residual of aerosol component approach (5.0% for MG and 5.7% for VW) is somewhat smaller than that of standard GRASP (6.6%). The next sentence is added to the manuscript: "In addition, the GRASP/Component approach produces almost the same average residual (2.4±0.9% for MG and 2.4±1.0% for VW) as that of the standard GRASP algorithm (2.3±0.9%) while the maximum residual for GRASP/Component (5.0% for MG and 5.7% for VW) is smaller than that for standard GRASP (6.6%); ± denotes standard deviation."

Figure R2. Inter-comparison of retrieval residual among different GRASP approaches. Red color represents the Maxwell-Garnett (MG) mixing model; blue - the volume-weighted (VW) mixing model; and black - the standard (ST) GRASP/PARASOL product that do not employ the aerosol component retrievals. The data subset used is the same as for the inter-comparison with the operational AERONET product (Table 4 and Figure

Regarding the used refractive indices: while addressing the first specific comment of the first reviewer, we added a table summarizing description of aerosol components and refractive indices (Table 2). In addition, an axis break is added for the ordinate in Figure 5a to better show the different range of real refractive indices. We also modified Figure 1 in the manuscript (see Figure R3 below).

Figure R3. General structure of GRASP algorithm with aerosol component conversion model, courtesy of (Dubovik et al., 2011). The red dashed frames represent modifications for the component inversion approach. fˆ* represents vector of inverted measurements, aˆP represents vector of unknowns at the p-th iteration, ãĂŰf(aãĂŮˆP) represents vector of measurement fit at the p-th iteration.

My final major comment is that there is really very little verification work provide that shows that the results of the retrievals are fundamentally better than other categorization methods (that is getting back to the question in point 2 (if this method leaves a residual from the free running GRASP algorithm). Or can you baseline against a simple AE vs AAE plot for species? One could argue that real verification has always been an issue for retrievals. There are many studies that show that DK retrievals provide reasonable results. But, those studies and here pick sites that are generally single aerosol specie dominated (the once exception is Solar village). This is why at least a self-consistent baseline against the standard GRASP and DK retrievals is so important. The discussions in Section 4 related to Figure 14-20 global maps that provide some truthiness. But close examination (which required me zooming way into the plots) shows many logical inconsistencies, especially around coastlines, where the overall hydration of the particles leads to an increase in the "soluble fraction" Likewise there is a great deal of "insoluble" AOD in Brazil-even though we know smoke organic components do in fact have a hygroscopicity to them, even if it is low), as well as retrieval errors. Are they not really just applying a form Schuster's "water fraction" algorithm? This then closes the loop with comment 1: are they really doing

a soluble and insoluble aerosol specie? In the end, I appreciate what the authors are trying to do, and can see how this will benefit the community. But there are logic issues that need to be wrung out, and some form of baseline verification that shows this method is actually taking us in the right direction. Response: The first part of this comment is indeed getting back to the question in point 2. Figure R2 and the discussion related to Figure 11 of the manuscript reply to this concern by showing the equivalence of the residuals and consistency of the retrievals between the new and the standard approaches. Regarding global maps, due to overall high complexity of the aimed approach, it is possible that not everywhere the retrieved aerosol components fit perfectly the expectations and a better understanding of what the retrieval represent is desirable. At the same time, not fully logical results are likely caused by a physical reason such as lack of sensitivity to some specific aerosol component, since optical observations in some cases sensitive to certain optical equivalent. This is challenge not only for the proposed methodology and for any other effort aimed to derive aerosol composition from remote sensing observations. In this respect, is can be noted that many known features are correctly represented in our retrievals. Moreover, some not yet known or fully understood phenomena can also be responsible for some "anomalies" and the proposed data set can be very useful basis for formulating new scientific question. Nevertheless, we agree that a more detailed and zoomed analysis should be in scope of future studies. However, due to very large volume of produced data in this publication and limited resources, we made the choice to present an overview of the results. Finally, we believe that the provided above and in the corrected version of the manuscript clarifications explain better the originality of the approach and the sense of the derived aerosol components.

Please also note the supplement to this comment:
https://www.atmos-chem-phys-discuss.net/acp-2019-208/acp-2019-208-AC2-supplement.pdf

[Figure]

[Figure]

**Fig. 1.** Figure R1. Water fractions for two-component aerosol mixtures as a function of the real refractive index. (Schuster et al., 2009)

**All sites**

[Figure]

**Fig. 2.** Figure R2. Inter-comparison of retrieval residual among different GRASP approaches.

ACPD
[Figure]

**Fig. 3.** Figure R3. General structure of GRASP algorithm with aerosol component conversion model, courtesy of (Dubovik et al., 2011).

[Figure]

---

## Referee Report (RR1)

[revised manuscript text omitted]

POLDER/PARASOL satellite observations.

[Figure]

[Figure]

**Figure 16.** Same as Fig. 15, but for BrC

[Figure]

**Figure 17**. Same as Fig. 15, but for fine mode non-absorbing soluble (FNAS)

[Figure]

[Figure]

**Figure 18**. Same as Fig. 15, but for coarse mode non-absorbing insoluble (CNAI, dust)

[Figure]

**Figure 19**. Same as Fig. 15, but for coarse mode absorbing insoluble (CAI, FeOx and Carbonaceous particles)

[Figure]

Figure 20. Same as Fig. 15, but for fine mode non-absorbing insoluble (FNAI, dust and OC)

**Response to comments of referee #1 on "Retrieval of aerosol composition directly from satellite and ground-based measurements" by Lei Li et al.**

We appreciate the referee's thoughtful reading, valuable suggestions and time that we hope helped us to improve the manuscript. Our point-to-point replies are presented below in blue.

**Anonymous Referee #1**

This paper presents an interesting strategy for using polarized/multiangular satellite measurements to infer aerosol composition. The method uses the Generalized Retrieval of Aerosol and Surface Properties (GRASP) along with assuming that aerosols are mixtures of non-soluble particles embedded within a soluble host. GRASP then derives size distribution, loading and light absorption characteristics based on determining the fractions of each aerosol type. The algorithm is applied first on synthetic data, then on ground-based AERONET, then on historic POLDER data.

**General comments:**

Although quite long, I find this paper to be well-organized, well-written and worthy of publication. I really like the idea that the GRASP retrieval can combine specific aerosol types (size + shape + absorption) to retrieve the aggregate. This makes it possible to compare directly with models (that also assume these types). I think the authors should highlight this even more than they have already. Finally, this is probably well beyond the scope of the paper, But I would like to see some sort of graphic (even for a single panels from Figs 15-19), to compare how your global maps (of species type) compare with other maps, such as from AeroCom models, and/or MISR size/shape/absorption climatology. Right now, they look reasonable, but I would be curious whether they might "change" our way of thinking about aerosol types distribution.

**Response:** We appreciate the referee's suggestion. The direct bridging to the models is highlighted now in the abstract and the discussion is extended in the manuscript. Also, in fact, a comparison of our GRASP retrievals with GEOS-5/GOCART model estimations was already done. However, as the referee mentioned, this next step is beyond the scope of the current publication. Either this observation-based aerosol components maps will "change" or not our way of thinking about aerosol types distribution is highly interesting and desires a more focused effort. Nevertheless, to give a glance on how the retrievals agree with the models, a couple of additional figures are presented below. Figure R1 shows the correlations for monthly averaged dust mass concentrations over the globe for 2008. It shows, for example that the global average coarse dust mass concentration retrieved by GRASP (125.5 mg/m$^2$) is lower than that simulated by GEOS-5/GOCART (174.7 mg/m$^2$). Figure R2 shows the map of coarse mode dust distribution in April 2008. The retrievals tend to provide lower coarse dust concentration over the mineral dust region (such as in Northern Africa and Northern China), which is consistent with the previous studies showing that models overestimate mineral dust in dust-dominated places (Jones and Christoper, 2007; Ginoux et al., 2001, Chin et al., 2009). Also, the observations show quite a fine spatial distribution structure.

[Figure]

[Figure]

**Figure R1.** Correlation between the monthly averaged dust mass concentrations of GRASP retrievals and GEOS-5/GOCART model estimations over the globe in 2008.

Dust mass concentration (GEOS–5/GOCART estimation)

[Figure]

Dust mass concentration (GRASP retrieval)

[Figure]

**Figure R2.** Distribution of dust mass concentration estimated by GEOS-5/GOCART model and retrieved by GRASP/POLDER in April 2008.

Chin, M., Diehl, T., Dubovik, O., Eck, T. F., Holben, B. N., Sinyuk, A., and Streets, D. G.: Light absorption by pollution, dust, and biomass burning aerosols: a global model study and evaluation with AERONET measurements, Ann. Geophys., 27, 3439-3464, 2009.

Ginoux, P. Chin, M., Tegen, I., Prospero, J. M., Holben, B., Dubovik, O., and Lin, S.-J.: Sources and distributions of dust aerosols simulated with the GOCART model, 106(D17), 20255-20273, 2001.

Jones, T. A. and Christopher, S. A: MODIS derived fine mode fraction characteristics of marine, dust, and anthropogenic aerosols over the ocean, constrained by GOCART, MOPITT, and TOMS, J. Geophys. Res., 112(D22), 2007.

**Specific comments: some small suggestions for improving the paper.**

1) Specifically, what are the aerosol type/model components? I feel as if a table could be used to describe each component, its size/shape and refractive index components.

**Response**: This useful suggestion is addressed by presenting a table summarizing the aerosol components description. The complex refractive indices of each component are presented at 0.440 $\mu m$ and 0.865 $\mu m$.

**Table 2.** Description of aerosol components and complex refractive indices at 0.440 $\mu m$ and 0.865 $\mu m$ employed in the GRASP components retrieval approach, as well as those used in the uncertainty tests.

| Abb. | Component | Complex refractive index | | Reference |
|------|-----------|-------------|-------------|-----------|
| | | 0.440 $\mu m$ | 0.865 $\mu m$ | |
| BC | Black carbon representing wavelength-independent strong absorption | 1.95+0.79i | 1.95+0.79i | Bond & Bergstrom (2006) |
| | | 1.75+0.63i | 1.75+0.63i | Bond & Bergstrom (2006) |
| BrC | Brown carbon representing wavelength-dependent absorption | 1.54+0.07i | 1.54+0.003i | Sun et al. (2007) |
| | | 1.54+0.06i | 1.54+0.0005i | Kirchstetter et al. (2004) |
| FNAI | Fine mode non-absorbing insoluble representing fine non-absorbing dust and organic carbon | 1.54+0.0005i | 1.52+0.0005i | Ghosh (1999) |
| | | 1.53+0.005i | 1.53+0.005i | "GKI"[1] |
| | | 1.52+0.0006i | 1.50+0.0006i | Koepke et al. (1997) |
| FNAS | Fine mode non-absorbing soluble representing inorganic salts | $1.337+10^{-9}$i | $1.339+10^{-8}$i | Tang et al. (1981); Gosse et al. (1997) for "AN"[2] |
| | | $1.537+10^{-7}$i | $1.517+10^{-7}$i | Toon et al. (1976) for "AS" [3] |
| FAWC | Fine mode aerosol water content | $1.337+10^{-9}$i | $1.329+10^{-6.5}$i | Hale & Querry (1973) |
| CAI | Coarse mode absorbing insoluble representing iron oxides | 2.90+0.345i | 2.75+0.003i | Longtin et al. (1988) |
| | | 2.88+0.987i | 2.72+0.140i | Triaud (2005) |
| CNAI | Coarse mode non-absorbing insoluble represented by non-absorbing dust | 1.54+0.0005i | 1.52+0.0005i | Ghosh (1999) |
| | | 1.53+0.005i | 1.53+0.005i | "GKI"[1] |
| CNAI | by Organic Carbon | 1.52+0.0006i | 1.50+0.0006i | Koepke et al. (1997) |
| CNAS | Coarse mode non-absorbing soluble represented by an inorganic salt - AN[2] | $1.337+10^{-9}$i | $1.339+10^{-8}$i | Tang et al. (1981); Gosse et al. (1997) |
| CNAS | by AS[3] | $1.537+10^{-7}$i | $1.517+10^{-7}$i | Toon et al. (1976) |
| CAWC | Coarse mode aerosol water content | $1.337+10^{-9}$i | $1.329+10^{-6.5}$i | Hale & Querry (1973) |

"GKI"[1] denotes dust composed of a mixture of quartz (Ghosh, 1999), kaolinite (Sokolik and Toon, 1999) and illite (Sokolik and Toon, 1999) with the proportions recalculated from Journet et al. (2014).

"AN"[2] denotes ammonium nitrate, which can be used to create a host in aerosols.

"AS"[3] denotes ammonium sulfate, which is an alternative species for the host estimation in aerosols.

2) Around line 290, the terms Frac(i) and dV/dlnr appear without definition. I guess they are in the table 1, but since table 1 (in the PDF) wasn't near the text, one might want to define first time in the text.

**Response**: Thank you for noticing this. We added the definitions of "$Frac(i)$ and $dV(r_j)/dlnr$ $(i = 1, ..., N_r)$" in the text.

3) The stars on equation (1) are confusing. Do the stars represent a priori or solutions?

**Response**: The term with a star represents satellite measurements. For example, $f_i^*$ denotes a vector of the measurements and $f_i$ denotes a vector of the estimations. It is now clarified in the text.

4) In line 355, Csph appears (see comment #2).

**Response**: The $C_{sph}$ definition was added in the text.

5) How are the intrinsic aerosol parameters allowed to vary in time and space?

**Response**: Besides the aerosol component retrieval module, the presented algorithmic developments essentially rely on the available heritage of the previous retrieval developments for POLDER space instrument. Thus, as in the standard GRASP algorithm (Dubovik et al., 2011), the satellite retrieval is designed as a statistically optimized simultaneous fitting of the observations over a group of pixels implemented under additional inter-pixel constraints. Specifically, the variations of the retrieved parameters horizontally from pixel-to-pixel or temporary from day-to-day over the same pixel are limited by the additional a priori constraints, in a similar manner to how it is applied in inverse modeling by Dubovik et al. (2008). The inclusion of these additional constraints is expected to provide retrieval of higher consistency for aerosol retrievals from satellites, because the retrieval over each single pixel will be benefiting from coincident aerosol information from neighboring pixels, in addition to the information about surface reflectance (over land) obtained in preceding and consequent observations over the same pixel.

Dubovik, O., Herman, M., Holdak, A., Lapyonok, T., Tanré, D., Deuzé, J.L., Ducos, F., Sinyuk, A., Lopatin, A.: Statistically optimized inversion algorithm for enhanced retrieval of aerosol properties from spectral multi-angle polarimetric satellite observations, Atmos. Meas. Tech., 4, 975–1018, https://doi.org/10.5194/amt-4-975-2011, 2011.

Dubovik, O., Lapyonok, T., Kaufman, Y.J., Chin, M., Ginoux, P., Kahn, R.A., Sinyuk, A.: Retrieving global aerosol sources from satellites using inverse modeling, Atmos. Chem. Phys., 8, 209–250, https://doi.org/10.5194/acp-8-209-2008, 2008.

6) For Eq (7), there appear to be a lot of zeros in the matrix. We can assume there is no covariance? For example, I can't see avc (volume size distribution) and av (total volume) as being independent

**Response**: Zeros in the matrix denote that there are no smoothness constraints for these parameters, such as the spectral smoothness constraint. A clarification sentence is added now in the text.

7) Line 396-397: I guess I am curious, what do you mean by: "choice of mixing rule…significantly affects the results". Can you show something about this?

**Response**: This statement should be indeed clarified. The meaning is that the different assumptions on mixing rules, which are often used for estimation of aerosol composition, were reported by Xie et al. (2014) as affecting the aerosol composition retrieved by an independent algorithm that uses ground-based AERONET measurements. For instance, the Bruggeman approximation was found as more suitable for the dust case, the Maxwell-Garnett for the haze case, and volume average for the clean case. In our study, the Maxwell-Garnett and volume average mixing rules were employed independently of the aerosol type and the retrieval results are inter-compared (Figure 11). In our approach we have not identified a significant influence of the mixing rule choice on the quality of the retrievals. Moreover, the aerosol optical properties were rather well comparable in both cases. The fractions of the elements present some differences due to the differences in the formulation, but are still in a reasonable agreement. The text is modified accordingly.

Xie Y., Li Z., Li L. et al.: Study on influence of different mixing rules on the aerosol components retrieval from ground-based remote sensing measurements. Atmospheric Research, 145: 267-278, 2014.

8) It appears that Eq (11) and Eq (12) have the same RHS?

**Response**: No, please note that there is a difference in the sign before $\varepsilon_r$.

9) Line 448 looks like a formula (11 minus 12) not 11 and 12.

**Response**: It is corrected.

10) Lines 455-484: See my comment #1.

**Response**: A table summarizing description of the aerosol components is added (see Table 2 in the revised manuscript and the response to comment #1).

11) I have a few comments regarding figure 5 and paper text. Could Fig 5a be split into two? There is a huge range of real refractive indices for CAI, but not for BrC and NAI. I cannot tell if the differences in assumptions for BrC and NAI around 1.5 are significant.

**Response**: An axis break is added for the ordinate in Figure 5a to better show the different range of real refractive indices.

12) Line 532-540. These range of values could also be added to a table (e.g. #1)

**Response**: The values of complex refractive indices at $0.440 \ \mu m$ and $0.865 \ \mu m$, used in the uncertainty test, are added to Table 2 (Please see the response to comment #1).

13) Line 545-546: "Elevated" meaning larger loadings or higher altitudes?

**Response**: It is reworded to "Therefore, the estimates should be quite reasonable in the cases of large pollution loading."

14) Lines 547-550: Are the fractions of BC and CAI somehow constrained so they can't be "large"? I note that they never approach 0.5 and barely approach 0.1

**Response**: It is important to say that the maximal retrieved fractions for BC and CAI

(mainly representing iron oxides) do not result from a constrain, despite have a limit due to the range of possible complex refractive indices in the pre-computed kernels of aerosol optical characteristics. That is, the volume fractions of these two highly absorbing species are indeed limited in the algorithm to 10% for BC and 3% for CAI, based on the below listed reasons. However, our analysis showed that these maximal values were never reached in the inversion procedure. Previous in-situ studies (Ganor and Foner, 1996; Guieu et al., 2002; Lafon et al., 2004, 2006, Alfaro et al., 2004; Wagner et al., 2012; Formenti et al., 2014) showed that iron oxides account for 2.8−6.5% of mineral dust by mass. It means approximately 1.4−3.25% by volume, since the density of free iron is much higher than other common minerals (4.28 g cm$^{-3}$ for goethite and 5.25 g cm$^{-3}$ for hematite, as opposed to 2.65 g cm$^{-3}$ for illite, kaolinite, quartz, and calcite; Formenti et al., 2014 The fraction of BC in atmospheric aerosol was generally reported not exceeding 10% (Bond et al., 2013).  It is also to note that the retrievals of aerosol composition derived from AERONET measurements by Schuster et al. (2016) demonstrated that the volume fraction of free iron remains relatively constant in West Africa throughout the year (1.4−1.7%) and the volume fraction of black carbon reaches a peak of 1.0% for the fine mode during West African biomass burning season and a peak of 3.0% for the fine mode in southern Africa biomass burning. The corresponding precision is added to the manuscript (lines 590 to 609).

Alfaro, S., Lafon, S., Rajot, J., Formenti, P., Gaudichet, A., and Maille, M.: Iron oxides and light absorption by pure desert dust: an experimental study, J. Geophys. Res., 109, D08208, doi:10.1029/2003JD004374, 2004.

Bond, T.C., Doherty, S.J., Fahey, D.W., Forster, P.M., Berntsen, T., Deangelo, B.J., Flanner, M.G., Ghan, S., Kärcher, B., Koch, D., Kinne, S., Kondo, Y., Quinn, P.K., Sarofim, M.C., Schultz, M.G., Schulz, M., Venkataraman, C., Zhang, H., Zhang, S., Bellouin, N., Guttikunda, S.K., Hopke, P.K., Jacobson, M.Z., Kaiser, J.W., Klimont, Z., Lohmann, U., Schwarz, J.P., Shindell, D., Storelvmo, T., Warren, S.G., Zender, C.S.: Bounding the role of black carbon in the climate system: A scientific assessment, J. Geophys. Res. Atmos., 118, 5380–5552, https://doi.org/10.1002/jgrd.50171, 2013.

Formenti, P., Caquineau, S., Chevaillier, S., Klaver, A., Desboeufs, K., Rajot, J.L., Belin, S., Briois, V.: Dominance of goethite over hematite in iron oxides of mineral dust from Western Africa: Quantitative partitioning by X-ray absorption spectroscopy, J. Geophys. Res. Atmos., 119, 12740-12754, https://doi.org/10.1002/2014JD021668, 2014.

Ganor, E. and Foner, H. A.: The mineralogical and chemical properties and the behavior of aeolian Saharan dust over Israel, in: The Impact of Desert Dust Across the Mediterranean, edited by: Guerzoni, S., and Chester, R., Kluwer Academic Publishers, Printed in the Netherlands, 163-172, 1996.

Guieu, C., Loye-Pilot, M. D., Ridame, C., Thomas, C.: Chemical characterization of the Saharan dust end-member: Some biogeochemical implications for the western Mediterranean Sea, J. Geophys. Res., 107 (D15), 4258, https://doi.org/4210.1029/2001JD000582, 2002.

Lafon, S., Rajot, J. L., Alfaro, S. C., Gaudichet, A.: Quantification of iron oxides in desert aerosol, Atmos. Environ., 38, 1211-1218, 2004.

Lafon, S., Sokolik, I.N., Rajot, J.L., Caquincau, S., Gaudichet, A.: Characterization of iron oxides in mineral dust aerosols: Implications for light absorption, J. Geophys. Res. Atmos., 111, 1–19, https://doi.org/10.1029/2005JD007016, 2006.

Schuster, G.L., Dubovik, O., Arola, A.: Remote sensing of soot carbon - Part 1: Distinguishing different absorbing aerosol species, Atmos. Chem. Phys., 16, 1565–1585, https://doi.org/10.5194/acp-16-1565-2016, 2016.

Wagner, R., Ajtai, T., Kandler, K., Lieke, K., Linke, C., Müller, T., Schnaiter, M., and Vragel, M.: Complex refractive indices of Saharan dust samples at visible and near UV wavelengths: a laboratory study, Atmos. Chem. Phys., 12, 2491–2512, doi:10.5194/acp-12-2491-2012, 2012.

15) For the plots of Figure 6, I am wondering what the "uncertainty" is. Should I read this as uncertainty is fraction (%) of fraction? What if these were presented in same units as x-axis (fraction)? Of course estimates of tiny fractions should have large % uncertainties, (but then that also means that the estimates of the fraction of the other elements will have lower % uncertainties).

**Response**: Thank you for this comment, it is indeed not sufficiently explained in the text. The uncertainty is defined in percentage as (retrieved_fractionassumed_fraction)/assumed_fraction. We would prefer to leave the notation in Figure 6 as it is, however, an explanation is added in the paragraph on uncertainty calculation (lines 646 to 647).

16) Line 581. I don't understand: "The non-absorbing insoluble can stand also for the insoluble organic carbon"

**Response**: The sentence is reworded. The intention was to be mentioned in the text problematic that the non-absorbing dust and non-absorbing organic carbon have similarity in the complex refractive index and is not distinguishable for the employed type of observations. It is corrected to "The non-absorbing insoluble component can represent not only non-absorbing dust, but also non-absorbing organic carbon, as was mentioned above."

17) Some of the figures have panels with cut-off axes (e.g. Fig 3)

**Response**: Corrected.

18) Are the units Fig 15-19 correct? ($mm^3/m^2$)?

**Response**: It can be confusing, but the units are correct. The units denote the volume concentration in total atmospheric column with unit surface area. We clarified it in the text.

**Anonymous Referee #2**

**General comments:**

This paper describes a modification to the GRASP algorithm to generate retrievals of a predefined basis set of aerosol "species." Typically the GRASP and the Dubovik and King (hereafter DK) provide a retrieval of size and real/complex refractive index. This is then used by others to classify species. Here they tie the refractive indices retrieved by GRASP by a basis set aerosol species. Their primary point is that using a predefined basis set of species is more direct than inferring typing from the measured extinction and subsequently derived absorption angstrom exponent method (or in a few cases adding index of refraction) commonly used in the community. As I finished this review, I did do a quick comparison with reviewer 1. I would agree with their point that perhaps the most important aspect of this work is it provides something of a forward operator to perform more apples to apples comparisons between satellite and models to close the radiance fields. I also agree that discussion of this point should be expanded. This discussion needs to cover these points. I think this effort moves the field forward in that they change the basis set as to what the retrieval is producing (BC, BrC, soluble and insoluble). However, these parameters are by no means equivalent to "composition," which is their premise for the entire paper. "Soluble and insoluble fractions" are wholly ill-defined in the context of composition, as they are related really to hygroscopicity. This is also true in part in regard to BC and BrC as they were functionally optical parameters long before we knew much of their true chemical nature. Here they are an indicator of spectral absorption properties. There are a myriad of soluble species with different indices of refraction and hygroscopic properties and likewise spectral dependencies of absorption based on mixtures. Thus, the idea that what is being retrieved as independent information on composition is fundamentally not true. What is a step forward, is they demonstrated that using the GRASP algorithm, you can generate a retrieval of parameters other than the standard size, index of refraction etc. as a basis of some categorization of the optical environment. One could look at this as a complex transform, but really it is simply a way of having an a priori set of basis functions for different aerosol species. But this is sort of what most traditional aerosol retrievals do, provide a best fit on the developers notion of what the aerosol environment looks like. Coarse mode dust, fine mode pollution, absorbing components, etc. So when the authors say they are the first ones to extract composition directly from satellite, this is not true either, it is just the first time in the GRASP algorithm has taken this approach. Rather, they are taking some liberty with the language. My first major comment is I think the authors need to be very careful about their language here. They should be more up front as to what they are doing, or spend much more time explaining why they think there is something fundamentally different in their approach.

**Response**: We appreciate this very interesting review and valuable thoughts that pushed us to better communicate the novelty and originality of the proposed approach. First of all, we understand the importance and sensitivity to the semantic and agree that the word "composition" can be more associated with aerosol chemical composition, which does not properly reflect what is retrieved. However, to define as retrieval of aerosol type or classification will be wrong as well, as explained below. We therefore converged to aerosol "component" that will be used hereafter.

We argue here and provide a discussion in the manuscript on why the suggested approach should be distinguished from the traditional aerosol retrievals (LUT approaches) and is fundamentally different to previous approaches. It should be noted that the retrieval of aerosol type has been clearly recognized as an important task by the scientific community and has been addressed in several studies. For example, there are a number of approaches that attempt to identify the type of aerosol through analysis of optical parameters such as single scattering albedo (SSA), Ångström Exponent (AE), AAE (absorption AE), refractive index, etc. Specifically, Russell et al. (2014) relate AERONET- and POLDER-derived optical properties to different aerosol types: urban, dust, marine, biomass burning, etc. Studies by Chung et al. (2010) and Bahadur et al. (2012) use AERONET optical properties like AE and AAE to separate BC, BrC, and dust into species-specific AAOT (absorption AOT). Schuster et al. (2005, 2009, 2016a) and Li et al. (2015) quantify the relative volume fractions of one or more aerosol species (e.g. BC, BrC, iron oxide, water) by adjusting the mixture of several components in an aerosol model to fit AERONET-retrieved refractive indices. However, our new approach differs substantially from all of these methods because it does not use a retrieval of optical parameters as an intermediate step. Thus, we expect the GRASP/Component approach to provide a stronger link to the radiation field than the previous approaches, as well as fundamentally higher retrieval accuracy.

Moreover, some of above methods have additional differences and limitations compared to our proposed approach. For example, the Russell et al. (2014) approach is rather qualitative and does not attempt to quantify the relative volume or mass fractions of different species in an aerosol mixture. Chung et al. (2012) and Bahadur et al. (2012) seem to use a technique for separating carbonaceous aerosols from dust that is not fully consistent with the AERONET retrieval assumptions, as discussed by Schuster et al. (2016b).

Also, the Look-Up Table (LUT) approaches employed in most satellite retrievals (Martonchik et al., 1998; Remer et al., 2005; Kahn and Gaitley, 2015; Popp et al., 2016; Hammer et al., 2018; etc.) are designed to search amongst a preselected set of aerosol models (or their mixtures) for a model that provides the best fit to the observations. Since the models in a LUT are usually associated with a number of aerosol types (e.g. desert dust, smoke, urban aerosol etc.), the identification of the model that provides the best fit is often considered as a retrieval of aerosol type/composition. For observations with enhanced sensitivity, such as the Multi-angle Imaging SpectroRadiometer (MISR), a large number of models can be justified in the LUT and the differentiation of the models described by the ensembles of parameters can indeed be rather robust. However, LUT approaches are fundamentally limited to a discrete set of possible solutions, whereas the GRASP/Component approach searches through a continuous space of solutions; thus, the identification of aerosol components with our new methodology is significantly more detailed and elaborate. The proposed approach also bridges directly to the quantities of aerosol compositions used in the global chemical transport models. Specifically, our aerosol component retrievals can satisfy the requirements of chemical transport models to constrain their aerosol estimations on a large or global scale. However, we note that the GRASP/Component approach is only possible if 1) there is significant instrument sensitivity to the parameters that are related aerosol component (i.e. complex refractive index), and 2) this sensitivity is maintained while other parameters like the size distribution are adjusted.

The retrieval provided in this article is unique because the aerosol component fractions are iterated in the original algorithm until a "best fit" is achieved with the measured radiance field, making it unnecessary to use retrieved optical properties as a constraint. Thus, the aerosol volume fractions determined with this new procedure have a direct link to the measured radiance field, whereas the procedures outlined above have a direct link to the retrieved aerosol optical properties provided by AERONET or satellite products.

In addition, the sensitivity of aerosol water fraction to the real refractive index is shown in Figure R1 (Schuster et al., 2009). Schuster et al. (2009) illustrated that the soluble aerosol components (sea salt, ammonium sulfate, ammonium nitrate, etc.) indicate similar refractive indices for similar mixing ratios, even though the dry refractive indices can be quite different. Hence, the aerosol water fraction can be derived from the mixture real refractive index if the aerosols are known to be one of the common soluble aerosols (in this study it is assumed as ammonium nitrate). Hygroscopicity, as an indicator of how aerosol growth in size or scattering, is related to relative humidity. However, in this study we are more concerned about how aerosol water content is related to real refractive index, which is much simpler (no hysteresis, no exponential growth, etc.) according to the study of Schuster et al. (2009) shown as Figure R1. These discussions are added to the manuscript.

[Figure]

**Figure R1.** Water fractions for two-component aerosol mixtures as a function of the real refractive index. (Schuster et al., 2009)

[revised manuscript text omitted]

My second point is that it is unclear as to what is really going on in the retrieval. If one considers the GRASP algorithm simply has a smoothness of fit contain, but otherwise can pick any refractive index and size it wants, why not just take the standard GRASP algorithm as it is, and after the fact match the size and refractive index to any basis function they like? Or is that what they are already doing? (it is hard to tell in figure 1 and the associated discussion). Is the goodness of fit always the same from the standard GRASP algorithm, or does your predefinition of species leave a residual? If so, how big is that residual? They also list as an example the refractive index for ammonium nitrate, but what of the other species? We are referred to a GACP dataset, and Figure 3 has such dynamic range between species it is hard to tell even what these values are. I think an appendix needs to be generated that provides details on these key aspects of the retrieval. I think in order for them to prove validity of the algorithm, they should do the retrieval with their basis set, and then as a baseline compare to the standard grasp algorithm, and see to what extent the goodness of fit to the radiance fields changes.

**Response**:

Regarding the goodness of fit and validity of the new algorithm as compared to the standard GRASP algorithm: we thank for bringing up this point because it is indeed an essential step that was done in early stages of the new algorithm developments, but is not explicitly mentioned in the original manuscript. Figure 11 of the original version, however, already showed an inter-comparison of correlations between optical characteristics derived by the new approach, the standard GRASP and the operational AERONET product, which addresses in a way the raised question. Here, in Figure R2, we present the residuals from different GRASP approaches. The data subset used is the same as for the inter-comparison with the operational AERONET product (Figure 11 and Table 4). Figure R2 shows that the residuals of the GRASP/Component approaches (Maxwell-Garnett (MG) and volume weighted (VW) mixing rules) are almost the same as those of standard GRASP. Although the average residual of aerosol component approach (2.4±0.9% for MG and 2.4±1.0% for VW) is slightly higher than of standard GRASP (2.3±0.9%), the maximum residual of aerosol component approach (5.0% for MG and 5.7% for VW) is somewhat smaller than that of standard GRASP (6.6%). The next sentence is added to the manuscript: "In addition, the GRASP/Component approach produces almost the same average residual (2.4±0.9% for MG and 2.4±1.0% for VW) as that of the standard GRASP algorithm (2.3±0.9%) while the maximum residual for GRASP/Component (5.0% for MG and 5.7% for VW) is smaller than that for standard GRASP (6.6%); ± denotes standard deviation."

[Figure]

**Figure R2.** Inter-comparison of retrieval residual among different GRASP approaches. Red color represents the Maxwell-Garnett (MG) mixing model; blue - the volume-weighted (VW) mixing model; and black - the standard (ST) GRASP/PARASOL product that do not employ the aerosol component retrievals. The data subset used is the same as for the inter-comparison with the operational AERONET product (Table 4 and Figure 11 of the manuscript).

Regarding the used refractive indices: while addressing the first specific comment of the first reviewer, we added a table summarizing description of aerosol components and refractive indices (Table 2). In addition, an axis break is added for the ordinate in Figure 5a to better show the different range of real refractive indices. We also modified Figure 1 in the manuscript (see Figure R3 below).

**General structure of inversion algorithm**

**Figure R3.** General structure of GRASP algorithm with aerosol component conversion model, courtesy of (Dubovik et al., 2011). The red dashed frames represent modifications for the component inversion approach. $f^*$ represents vector of inverted measurements, $a^P$ represents vector of unknowns at the $p$-th iteration, $f(a^P)$ represents vector of measurement fit at the $p$-th iteration.

My final major comment is that there is really very little verification work provide that shows that the results of the retrievals are fundamentally better than other categorization methods (that is getting back to the question in point 2 (if this method leaves a residual from the free running GRASP algorithm). Or can you baseline against a simple AE vs AAE plot for species? One could argue that real verification has always been an issue for retrievals. There are many studies that show that DK retrievals provide reasonable results. But, those studies and here pick sites that are generally single aerosol specie dominated (the once exception is Solar village). This is why at least a self-consistent baseline against the standard GRASP and DK retrievals is so important. The discussions in Section 4 related to Figure 14-20 global maps that provide some truthiness. But close examination (which required me zooming way into the plots) shows many logical inconsistencies, especially around coastlines, where the overall hydration of the particles leads to an increase in the "soluble fraction" Likewise there is a great deal of "insoluble" AOD in Brazil-even though we know smoke organic components do in fact have a hygroscopicity to them, even if it is low), as well as retrieval errors. Are they not really just applying a form Schuster's "water fraction" algorithm? This then closes the loop with comment 1: are they really doing a soluble and insoluble aerosol specie? In the end, I appreciate what the authors are trying to do, and can see how this will benefit the community. But there are logic issues that need to be wrung out, and some form of baseline verification that shows this method is actually taking us in the right direction.

**Response**: The first part of this comment is indeed getting back to the question in point 2. Figure R2 and the discussion related to Figure 11 of the manuscript reply to this concern by showing the equivalence of the residuals and consistency of the retrievals between the new and the standard approaches. Regarding global maps, due to overall high complexity of the aimed approach, it is possible that not everywhere the retrieved aerosol components fit perfectly the expectations and a better understanding of what the retrieval represent is desirable. At the same time, not fully logical results are likely caused by a physical reason such as lack of sensitivity to some specific aerosol component, since optical observations in some cases sensitive to certain optical equivalent. This is challenge not only for the proposed methodology and for any other effort aimed to derive aerosol composition from remote sensing observations. In this respect, is can be noted that many known features are correctly represented in our retrievals. Moreover, some not yet known or fully understood phenomena can also be responsible for some "anomalies" and the proposed data set can be very useful basis for formulating new scientific question. Nevertheless, we agree that a more detailed and zoomed analysis should be in scope of future studies. However, due to very large volume of produced data in this publication and limited resources, we made the choice to present an overview of the results. Finally, we believe that the provided above and in the corrected version of the manuscript clarifications explain better the originality of the approach and the sense of the derived aerosol components.

---

## Author Response (AR2)

**Response to comments on "Retrieval of aerosol composition directly from satellite and ground-based measurements" by Lei Li et al.**

We appreciate the referee's time, constrictive and valuable comments that helped us to improve the manuscript. Our point-by-point responses are presented below in blue.

**Referee #3**

**General comments:**

1. First was a positive statement:

"the most important aspect of this work is it provides something of a forward operator to perform more apples to apples comparisons between satellite and models to close the radiance fields." I also agree that discussion of this point should be expanded. This is the most important aspect of the work and it is hidden in a single statement on lines 268-271 that I've put into highlight in my comments. This really is the main contribution of this method and paper. I actually came to this on my own after reading lines 268-271, without understanding the previous reviewers' comments. So, now you have it. Three independent reviewers all see the same thing. The main contribution the GRASP/Component retrieval makes is that it retrieves parameters that match models. This ability supercedes all the nuances of avoiding working from retrieved optical properties (which MISR does anyway) and LUTs. Thus, the authors' response to this comment is barely satisfactory.

**Response:** (included in the text) We appreciate the referee's comments. These useful suggestions are addressed by adding some descriptions in the revised version to emphasize the contribution of our method as "The proposed methodology eliminates two intermediate steps commonly used in comparing satellite results with models. First, the retrieval is done directly from measured radiances to components without passing through retrieved optical properties or assume aerosol models. Second, the methodology allows matching CTM components without passing through often ambiguous AOT to mass conversions, e.g. using like mass extinction efficiency."

2. Then a serious criticism:

"However, these parameters are by no means equivalent to "composition," which is their premise for the entire paper". I warned the authors about this very issue when I

first encountered this work as a Ph.D. dissertation over a year ago. There is no excellent solution to the semantics, but composition is definitely wrong. Component is not perfect, but it works. Thus, the authors' response to this comment is good.

**Response:** We appreciate the referee's comments very much, all corrections were accepted.

3. Next, a suggestion:

"They should be more up front as to what they are doing, or spend much more time explaining why they think there is something fundamentally different in their approach". The authors responded to this comment very well. The additional text is much welcomed and adds much to the paper.

I do have a problem with "a first attempt to derive aerosol components from satellite". Component is not much different from type. I don't see this work as fundamentally different from MISR. What might be said instead: "a first attempt to derive aerosol components without Look-Up Tables from satellites" OR "a first attempt at this more realistic method to derive aerosol components from satellite" OR "a first attempt to derive aerosol components linked to their hygroscopicity from satellite" OR "a first attempt to derive aerosol components from satellite specifically tied to global chemical transport model quantities" OR "a first attempt to derive aerosol components in this manner". I happen to like the fourth statement best.

**Response:** We appreciate the referee's comments. For the text, we also selected the fourth choice in the revised version as "a first attempt to derive aerosol components from satellite specifically tied to global chemical transport model quantities."

4. A question of not understanding what is going on in the algorithm:

"If one considers the GRASP algorithm simply has a smoothness of fit contain, but otherwise can pick any refractive index and size it wants, why not just take the standard GRASP algorithm as it is, and after the fact match the size and refractive index to any basis function they like?" The authors don't really address this in the text, as far as I can tell, except for updating the Figure. I like the new figure, and I seem to understand what is going on. I think. But obviously an informed reader (Reviewer #2) missed some key things. It might be worthwhile to add a simple statement at the beginning of the methodology to answer this question directly.

Authors response: Not satisfactory.

**Response:** (included in the text) At the beginning of the methodology, a statement was added to answer the question directly as "It was demonstrated that GRASP algorithm can provide retrieval of rather complete set of aerosol parameters from multi-angular satellite polarimetry or AERONET like ground-based observations. The set of retrieved parameters includes size distribution, complex refractive index, information about particle shape, as well as, various optical properties as AOT, single scattering albedo, Ångström Exponent, etc. The values of these parameters certainly are related to aerosol type, however, they do not provide any direct information about possible aerosol type or quantitative indication of various components presence in the observed aerosol. The identification of presenting aerosol types can be obtained by matching the retrieved parameters into preselected the basis of aerosol components with known properties (e.g., Russell et al., 2014). However, such approach relies on several intermediate steps based of the assumptions of rather different methodological nature and therefore each step introduces additional uncertainties and even ambiguity. The essence of methodological developments in this study is to develop an approach of retrieving directly aerosol components from observations with no intermediate steps and uniquely set assumptions."

5. Another question concerning the algorithm:

"Is the goodness of fit always the same from the standard GRASP algorithm, or does your predefinition of species leave a residual? If so, how big is that residual?"

Here the authors dig in, provide a good response, and add some key information in the form of a few summary sentences in the text. One thing to think about is that I found figure R2 more informative than the scatter plots that are found in the manuscript itself. The authors may want to add R2 to the Supplementary material.

Authors response: Very good.

**Response:** We added the figure R2 in the revised version (see Fig. 13 in revised version) to provide more information.

[Figure]

**Figure 13.** Inter-comparison of retrieval residuals among different GRASP approaches. Red color represents the Maxwell-Garnett (MG) mixing model; blue represents the volume-weighted (VW) mixing model; and black represents the standard (ST) GRASP/PARASOL product that do not employ the aerosol component retrievals.

6. The issue of verification:

"there is really very little verification work provide that shows that the results of the retrievals are fundamentally better than other categorization methods (that is getting back to the question in point 2 (if this method leaves a residual from the free running GRASP algorithm))."

The fact is that it is extremely difficult to verify components. Still, whatever is shown in the paper only tells us that the component method is "as good" as standard GRASP, not "better", in the parameters that standard GRASP derives. I think that is ok. But it would have been nice to have a solid statement in the text outlining the verification strategy. Maybe saying that the goal was to match standard GRASP in terms of AOD, AE and SSA, which gives support to the component divisions, but doesn't validate them. I think would satisfy Reviewer #2 better than what was said.

Response to review: Not satisfactory.

**Response:** We added in the some discussions to describe the verification strategy in the revised version as "The overall strategy of these tests, as well as of analysis of obtained results below in the paper, was, first, to demonstrate and verify that GRASP/Component approach matches the standard GRASP retrievals in terms of AOT, AE and SSA under the predefined component divisions. Then, second, if this match is achieved, to demonstrate the possibility of unique distinction of component in sensitivity tests and to verify agreement with available independent data of the retrieved results."

**Minor corrections**

There are 40 comments in the comments list. Most of these are minor editorial fixes, like typos. Please note that I changed "component" when it is a noun to "components" (plural), but left "component" (singular) when it is used as an adjective.

**Response:** We appreciate the referee's valuable suggestions and corrections that help us to improve the manuscript. We included all editorial suggestions in the revised version. For example, we changed "assumed as" into "assumed to be" in Line 35. We also changed "modelers community" into "the in situ and modeling community" and added a reference Schmeisser et al. (2017) in Line 844-845. Please see the details in the revised version.

[revised manuscript text omitted]

**General structure of inversion algorithm**

*Observation definition*

**Forward model:**
Simulates observations $f(a^p)$ for a given set of parameters $a^p$

*Replace aerosol refractive indices by refractive indices driven by fractions of aerosol components*

$a^p$    $f(a^p)$, etc.

**Observation definition:**
Viewing geometry, spectral characteristics; coordinates, etc.

**Observations:** $f^*$

$f^*$

**Numerical Inversion:**
Stat. optimized fitting of $f^*$ by $f(a^p)$ under a priori constraints

**Inversion settings:**
- description of error $\Delta f^*$;
- a priori constraints

$a^p$ - final

**Retrieved parameters:**
$a^p$ –describes optical properties of aerosol and surface

*- including refractive indices driven by aerosol components that directly retrieved from satellite observations by fitting radiation*

[revised manuscript text omitted]